# ALPHAMOL: RIGID BODY REPRESENTATION OF MOLECULAR STRUCTURES FOR PREDICTION OF SMALL MOLECULE GROUND-STATE

## ABSTRACT

Recent success of AlphaFold2 (Jumper *et al.*, 2021) in predicting structures of proteins from multiple sequence alignments (MSA) raises the question: can we generalize this approach to other important types of molecules? The positive answer to this question opens a door to overcoming the lack of structural data needed to train the model for predicting structures of RNA, proteins with non-standard amino-acids and proteins with post-translational modifications. In this work we presented a new model for predicting molecular structures, that generalizes AlphaFold2 approach to predicting structures of proteins. Two key contributions this work provides is a new representation of molecules as a collection of neighborhoods that behave as rigid bodies and a way to encode the bonds between rigid bodies into a prediction algorithm. We test this approach on the task of predicting ground-state structures of small molecules. Code available at AlphaMol repository.

## 1 INTRODUCTION

The overarching goal of computational structural biology is to develop precise predictive models capable of elucidating all chemical and structural transformations occurring within a cell. Similarly, the ultimate goal of machine learning is the development of a generally intelligent model capable of operating with comprehensive and various modalities of physical reality. The cutting edge research is currently centered on the creation of multi-modal large language models, leveraging diverse data sources such as audio, images, or even active robotic arm manipulators Driess *et al.* (2023); Huang *et al.* (2023). These seminal studies underscore the remarkable capacity of models trained on cross-modal datasets to transfer knowledge to novel tasks. Over the past decade, the application of machine learning has driven significant advancements in computational structural biology. Notably, the recent breakthrough in protein structure prediction by AlphaFold2 Jumper *et al.* (2021) is based on the transformer architectures which have emerged as dominant architectures in deep learning. Subsequent research focused on refining and extending AlphaFold2's capabilities, for example, predicting of protein-protein complexes Evans *et al.* (2021), utilizing single sequences as inputs instead of multiple sequence alignments (MSA) Lin *et al.* (2022), and implementing diffusion-based learning Ingraham *et al.* (2022); Watson *et al.* (2022). Moreover, recent successes in predicting the structures of non-coding RNA Pearce *et al.* (2022) and elucidating protein-ligand interactions based on AlphaFold2 descriptors Hekkelman *et al.* (2023) highlight the possibility of constructing a multi-modal model that works across various molecular domains. Given physico-chemical diversity of macromolecules (proteins, nucleic acids) as well as small molecules, the researches have also attempted to unify atomic description of aminoacid residues, drug-like organic molecules (ligands), nucleic acids and non-standard residues Krishna *et al.* (2023). However, in practice, protein and nucleic acid representations are typically retained as sequences of tokens, while ligands and non-standard amino acids are represented as atom graphs. This problem forced the authors to perform atomization of residue tokens by representing aminoacids or nucleotides as ligands. On the other hand, one of the remarkable features of AlphaFold2 is the coarse-grain protein representation, where each amino acid residue is described as a rigid body, while the inner degrees of freedom are predicted separately. This coarse-graining method allowed extracting the orientation of aminoacid residues from MSA, that plays a big role during the co-evolution of the neighboring protein residues, and used in the structural module of AlphaFold2 to predict structures more accurately compared to the

other methods. In this work we provide foundation for consistent model architecture that allow one to eliminate token-based representation of polymers, including proteins and nucleic acids. We proposed a novel representation of an arbitrary type molecule as a set of rigid bodies with additional constraints and demonstrated the utility of this representation by considering one of the most complex chemical type, namely, small molecular structures. We further improved the Evoformer block architecture from AlphaFold2, such that it incorporates explicit positions of rigid bodies eliminating the need for a separate structural module. We have tested our method on the task of predicting ground-state molecular structures using the curated dataset Molecule3d Xu *et al.* (2021c) extracted from PubChemQC Nakata and Shimazaki (2017) of $4 \cdot 10^6$ molecules. Our model achieves the average root mean square deviation (RMSD) of 0.83 over $7 \cdot 10^5$ molecules in the testing subset of the Molecule3d dataset, outperforming RDKit Landrum (2020) and other published methods. More specifically, the mean average error of the predicted distance matrix between the heavy atoms of the molecules is 0.30 versus 0.53 for the RDKit ETKDG algorithm Riniker and Landrum (2015) and 0.66 for the DeeperGCN-DAGNN + DistanceXu *et al.* (2021c).

## 1.1 PREVIOUS WORK

Existing methods to predict small molecule structures typically rely on graph-based methods, which treat molecules as a graph of nodes (atoms) connected by edges (bonds). One of the first such works is CVGAE Mansimov *et al.* (2019), however, the quality of structure predictions was impractical and required further optimization using molecular dynamics force fields. Another work, GraphDGSimm and Hernández-Lobato (2019), extends the molecular graph to the second and third atomic neighbors and transforms the graph into a distance matrix. Such graph extension was needed to fix the dihedral angles in the molecule. The prediction target of this method is the extended distance matrix of the conformer, and the 3D structure is then obtained by a non-differentiable Euclidian distance geometry method (EDG, Riniker and Landrum (2015)). This method relies on a standard message-passing neural network in conjunction with a conditional variational autoencoder and works entirely in the internal coordinates of a molecule. Thus, it generates predictions that are invariant with respect to rotation and translation. The following work CGCFXu *et al.* (2021b) improved training by using energy-based learning and neural ordinary differential equation approach, while using the same molecular representation. The next improvement, ConfVAE Xu *et al.* (2021a), was achieved by allowing gradient propagation through the EDG step using bilevel programming approach. This was the first method that allowed end-to-end training coupled with translational and rotational invariance. The ConfGF method Shi *et al.* (2021) utilized a different approach: they estimate the gradient of the logarithm of the probability distribution of the interatomic distances using an energy-based method and then derive an atomic gradient field using the chain rule. This work proposed the first equivariant end-to-end differentiable structure prediction algorithm for small molecules. However, because of the difficulties in training energy-based models, one forced to additionally sample structures perturbed using Gaussian noise. The other approach for end-to-end differentiable molecular structure prediction, GeomolGanea *et al.* (2021), directly predicts local neighborhoods of atoms and then assembles the whole structure by predicting torsion angles along the shared bonds between neighborhoods. Of note, this was the first method, that accounted for enantiomers of a molecule and still invariant with respect to the rotations and translations. One of the most recent methods, DMCG Zhu *et al.* (2022), that directly predicts the atomic coordinates, was the first to consider molecular graph isomorphisms in the loss function. Another branch of methods rely on diffusion-based learning to directly generate coordinates of atoms in molecules started by GeoDiffXu *et al.* (2022). SDEGen Zhang *et al.* (2023) applies diffusion to the distance matrices and Jing *et al.* (2022) in the torsional angle space. Finally, EC-Conf Fan *et al.* (2023) improves the sampling efficiency of the diffusion method for this task. Despite the considerable efforts that went into this field of machine learning, it is important to note, that the utility of the methods and evaluation metrics are still questionable. For example, Gengmo Zhou Zhou *et al.* (2023) and coauthors showed that the standard EDKTG algorithm implemented in RDkit with minor modifications outperforms all the previously mentioned methods on the GEOM-QM9 and GEOM-DRUGS datasets. Appendix Table 2 provides a list of the previously developed methods for small molecular structure prediction.

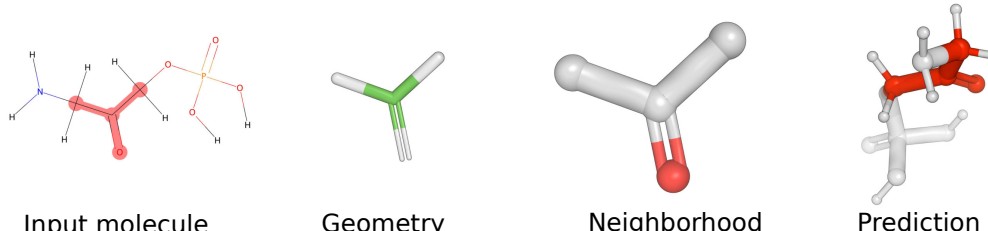

Input molecule          Geometry          Neighborhood          Prediction

Figure 1: The outline of our approach to the problem of molecular structure prediction. For each atomic neighborhood in a molecule, we construct a rigid body, that corresponds to the fixed bond geometry and inter-atomic bond distances. The model then predicts rotations and translations of neighborhoods.

## 2 METHODS

This section is organized as follows. First, we introduce one of the key distinctions of our approach, which is representation of a molecule as a set of rigid neighborhoods. Then we describe featurization of this molecular representation. This is followed by addressing the symmetry and chirality problem in prediction of spatial transforms of rigid bodies. Next, we touch on the equivariance of the spatial transforms predictions. Finally, we describe the model architecture, introducing a modified Evoformer block, and the loss functions.

### 2.1 MOLECULE REPRESENTATION

The idea is to decompose the input molecule as a set of atomic neigborhoods, and for each atomic neighborhood to use a predefined bond geometry and atomic distances, such that one can explicitly compute the atomic coordinates of a neighborhood. Given this representation, the model is trained to predict the rotation and translation of the whole neighborhood by matching the correct structure. Figure 1 shows an example of a rigid neighborhood. To construct the rigid bodies constituting the atomic neighborhoods we develop an algorithm that combine interatomic distances and geometries (see Appendix section A.1.), resulting in reconstruction of molecular neighborhoods with acceptable precision (Figure 1). We observed that local bond geometries can be rigorously approximated by the rigid bodies, except for some isolated cases (see Appendix Table 6). We classified bond geometries based on the hybridization of valent electron orbitals in the central atom and a set of bonds between the central atom and its neighbors. This entails that each geometry has a symmetry, which we compute by enumerating permutations of bonds and realigning them to the initial geometry. We observed, that only 22 different geometries are enough to cover most of the molecules in the Molecule3D dataset. Table 6 shows examples of the extracted arrangements and the RMSD distributions for the molecules from the dataset. Although, for some molecular structures neighborhoods deviate from the standard 22 geometries and average atomic distances (see Appendix Figure 10, row 4), these examples are rare and the algorithm is able to correct such small deviations (See Appendix section A.2.4).

### 2.2 FEATURES

Similarly to the AlphaFold2 model, the features we passed into the model are divided into single and pairwise ones. Single features encode each rigid body separately, while pairwise features encode spatial relationships between the rigid bodies. The algorithm to calculate the features is listed in Appendix Table 8. Each neighborhood is described by the ordered set of vectors corresponding to the concatenated central atom features, bond features, neighbor atom features, and neighbor atom coordinates in the neighborhood. We embed each neighborhood using a 4-layer fully connected neural network. Note, that the pairwise features consist of shared bond features between the two neighborhoods and alignment matrices and translations between them. The necessity of including specific transformation between the pairs of molecular neighborhoods is illustrated in Figure 2. If the geometric information about the specific bond shared between the two neighborhoods is not passed to the model, it leaves room for ambiguity and degrades model predictions. To obtain the alignment matrix $R_{ij}$ from a neighborhood $i$ to the neighborhood $j$, we compute the matrices $H_{ij}$ that align the

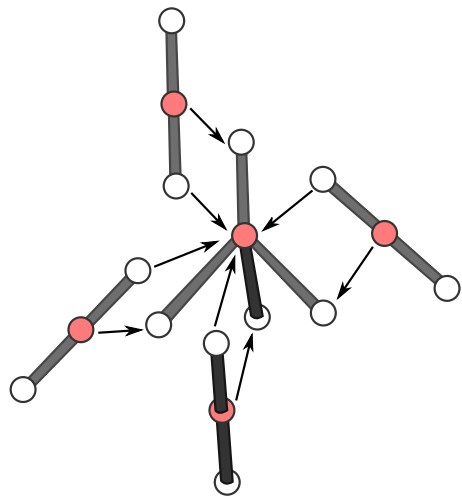

Figure 3: Example of an isomorphism of a molecule. Red and yellow highlight atoms swapping which the molecule graph does not change.

Figure 2: Example of some of the possible pairings between two neighborhoods. Red circles are the neighborhood central atoms.

shared bond parallel to the x-axis. The alignment matrix for neighborhood $i$ along the shared bond of neighborhood $j$ is then:

$$R_{ij} = H_{ji}^T H_{flip} H_{ij} \tag{1}$$

, where $H_{flip}$ is the matrix that flips x-axis. Similarly, we compute alignment translations $t_{ij}$ between the neighborhood pairs. These features depend only on the bond geometries and distances described in the section 2.1. Appendix section A.2 provides detailed description of the data processing pipeline and Appendix algorithms 9 and 10 shows the computation of the $R_{ij}$ matrices and $t_{ij}$ vectors.

## 2.3 SYMMETRY AND CHIRALITY

To predict rotations and translations of rigid bodies corresponding to a molecule, one has to take into account permutation symmetries of the molecule graph. Figure 3 shows a molecule, where swapping atoms 1,2,13,14 with the corresponding atoms 5,4,7,6 does not change the molecular graph. Therefore, for example the neighborhood comprisins C:2, O:13, C:1, and C:3 may correspond to two different rotations and translations. During the training process of our model, we generate all possible isomorphisms and then compute different combinations of target rotations and translations of neighborhoods.

The chirality of molecules strongly affects their properties and their interactions with biological molecules, therefore it has to be addressed carefully. In our case the same bond geometries with two different chiralities do not align. Therefore one needs to specify chirality of each neighborhood to get the correct bond geometry for prediction. During the training, we extract this data from the target structures using standard RDKit utilities. First, we assign ranks to atoms using CIP rules Cahn *et al.* (1966). Then for each neighborhood central atom, we arrange its neighbors in the order defined by the atomic rank (see Appendix algorithm 6). We assign a chiral label as the sign of the volume built on the vectors, formed by the bonds from the central atom of a neighborhood. During the extraction of the bond geometries, we verify that all geometries and allowed permutations have the positive chiral label (see Appendix algorithm 2). During the construction of the neighborhoods (see in Figure 1), we reflect the bonds with respect to the central atom, if the chirality of the neighborhood does not correspond to the label assigned by the RDKit (see Appendix algorithms 9 and 11). Note, that for some molecules, an isomorphism can swap atoms of a chiral center. If such permutation changes the chirality, we can choose it arbitrarily at the cost of the isomorphism. In this case, we fix the chirality and choose all the isomorphisms for which the neighborhood aligns with the ground truth structure (see Appendix algorithm 11). Also note, that for some molecules the number of isomorphisms is exceedingly large to store the coordinates of atoms for each one of them. Therefore, we factorize

isomorphisms into local and global ones, where local isomorphisms act only on one neighborhood, leaving every other atom in place (see Appendix section A.2.2).

## 2.4 EQUIVARIANCE

The rotations and translations predicted by the model should be equivariant, that is they should rotate or translate when the input is rotated or translated accordingly, formally written as:

$$F(g \circ x) = r \circ F(x)$$

where $F$ is the prediction model, $g$ and $r$ are the elements of the SE(3) group: the group of all rotations and translations of the 3-dimensional space. Here, we used a simplified definition of the equivariance, where $g = r$ and used rotation matrices and translation vectors as inputs and outputs.

Previous approaches to the construction of equivariant neural networks can be broadly separated in two classes: generally or specific equivariant networks. The first class of approaches can represent any $F$ that can be learned by a neural network. The methods belonging to this class either represented tensors as a decomposition into spherical harmonics Thomas *et al.* (2018) or require lifting $R^3$ space into high-dimensional Lie group space Hutchinson *et al.* (2021). Both of these approaches require recomputing decomposition into spherical harmonics which can be numerically unstable and computationally costly. The second class of approaches to equivariance is exemplified by the SchNet Schütt *et al.* (2017) and AlphaFold2 structure module. These methods include equivariant operations in the initial $R^3$ space into the model, which limits the function $F$ such a model can learn. Recently, a new approach emerged in the class of generally equivariant neural networks Du *et al.* (2022a); Wang and Zhang (2022), where the key idea is to construct a set of reference frames and project input tensors onto them. The predictions are then given as a set of decomposition coefficients that are transformed back into tensors. The seminal work by Du et. al. Du *et al.* (2022a) dealt with systems composed of sets of points, thus, the frame construction may lead to numerical instability in case of points being close in space or belonging to the same plane. Here, we work with a system comprised of rigid bodies with natural frames of reference associated with them, therefore avoiding the aforementioned problem.

Let $X = (R_1, T_1, ... R_N, T_N) \in R^{12N}$ be a many-body system embedded into $R^3$ space, where $N$ is the number of rigid bodies. For rigid body $i$, we use $R_i(t) \in R^9$ and $T_i(t) \in R^3$ to denote its rotation and translation at iteration $t$, respectively. The rows of the matrix $B_i^k = R_i = (a_i^1, a_i^2, a_i^3) \in R^3$ correspond to the basis vectors of each local frame associated with the neighborhood $i$ (these local frames make a complete basis for vectors). Additionally, we construct the basis for rotation matrixes using the approach outlined by Du et. al. Du *et al.* (2022a), namely $B_{ij}^{kl} = a_i^k \otimes a_j^l \in R^9$, where $k, l \in 1, 2, 3$ and $i, j \in 1 \dots N$. To preserve translation equivariance we first centralize the translations $T_i^{(c)} = T_i - \frac{1}{N} \sum_i^N T_i$. Then we take each vector input for the rigid body $i$ (like $T_i^{(c)}$) and compute its decomposition coefficients in the basis $B_i^k$. Similarly, for the input rotation matrixes defined on the pairs of rigid bodies (see $R_{ij}$ in section 2.2) we use a basis composed of matrices $B_{ij}^{kl}$. Additionally, we have input vectors defined on the pairs of neighborhoods ($t_{ij}$ see section 2.2), that we have to scalarize. To avoid constructing a new basis we instead transform these vectors into tensors: $t_{ij}^{(t)} = t_{ij} \otimes t_{ij}$ and then decompose them into basis $B_{ij}^{kl}$. The resulting scalars then can be used in a neural network without breaking the SE(3)-equivariance.

To obtain updated rotations and translations of the neighborhoods we predict translation and rotation update vectors and matrices, correspondingly. For the vector prediction, we interpret predictions as the decomposition coefficients in the basis $B_i^k$. Thus, translations updates are computed as $dT_i = \sum_k F_{ik} B_i^k$, where $F_{ik}$ is the $k$-th network output for $i$-th neighborhood. In principle, one can use the same vectorization procedure for computing rotation matrices updates using the basis $B_{ij}^{kl}$. However, in our case, the space of all possible rotation matrices forms a manifold in the space of all possible decompositions in the basis $B_{ij}^{kl}$. On the other hand, we need to predict only a valid rotation matrix, that is orthogonal matrix. For this, we use the 6D-rotation representation first proposed in the work Zhou *et al.* (2019), where the idea is that any rotation matrix can be represented using just two vectors $(b_1, b_2)$ and to construct the complete orthogonal matrix from these vectors one follows the

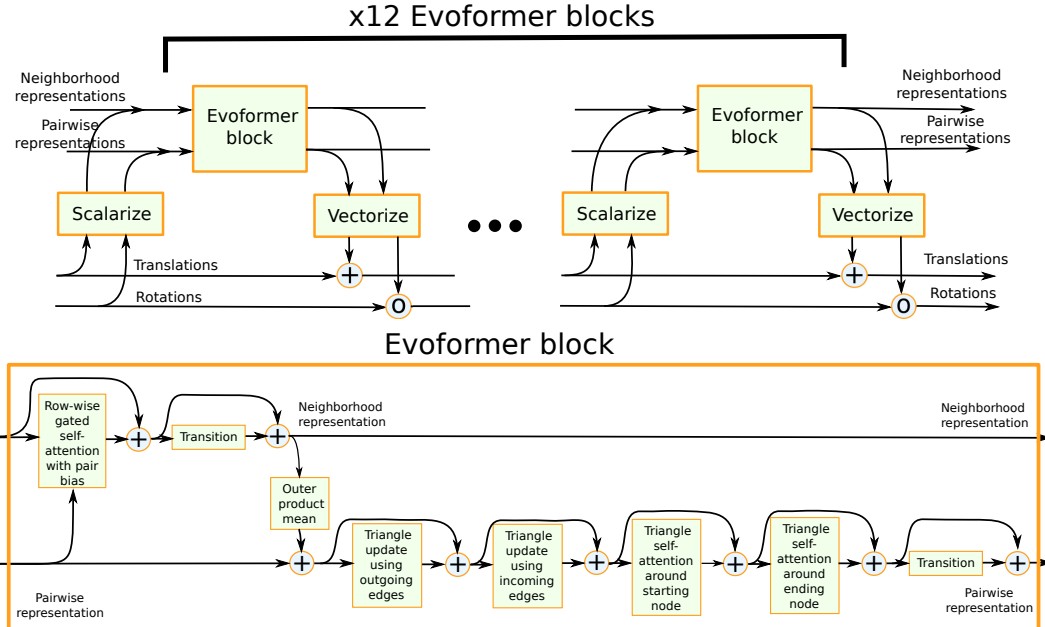

Figure 4: Schematic representation of the AlphaMol model along with the modified Evoformer block. Scalarization and vectorization blocks are described in section 2.4. "+" symbol denotes component-wise sum of feature tensors and "o" symbol denotes matrix multiplication.

Graham-Schmidt process:

$$GS\left(\begin{bmatrix} | & | \\ b_1 & b_2 \\ | & | \end{bmatrix}\right) = \begin{bmatrix} | & | & | \\ c_1 & c_2 & c_3 \\ | & | & | \end{bmatrix} = \begin{cases} c_1 = N(b_1) \\ c_2 = N(b_2 - (c_1 \cdot b_2)c_1) \\ c_3 = c_1 \times c_2 \end{cases} \tag{2}$$

, where $N$ is the vector normalization operation. Using this rotation representation we need to predict two vectors instead of a matrix, so we can use the same basis $B_i^k$, that we employed for predicting translation updates. The detailed description of vectorization and scalarization algorithms developed in this work presented in Appendix section A.3, particularly, Appendix algorithms 15 - 18.

## 2.5 MODEL

The model consists of sequential Evoformer blocks that predict rotations and translations updates to the initial rotations and translations of the molecular neighborhoods (see Figure 4). Each updated transform is computed as follows:

$$T_i = T_i^{pred} + T_i^{prev} \tag{3}$$

$$R_i = R_i^{pred} \circ R_i^{prev} \tag{4}$$

, where subscripts $pred$ and $prev$ corresponds to predicted transforms from each Evoformer module and previous total transform, respectively; and $i$ denotes the neighborhood index. We used identity matrices and zero vectors as the initial ones. Note, that the molecular representation comprises pairwise features to encode relations between the rigid neighborhoods and the single features to encode characteristics of the rigid neighborhoods itself. Therefore, the operations within the Evoformer block, treating the prediction of molecular structures as a graph inference problem in $R^3$, must reflect chemically feasible molecular geometries in the $R^3$ space. We would like to note, that operations on the pairwise representations are one of the key innovations introduced in AlphaFold2: they are based on the intuition, that the pairwise representations contain information that must satisfy the triangle inequality on distances. Therefore the corresponding update functions operate on triangles of edges involving three different nodes; the missing edge of the triangle is included using logit bias to axial attention coupled with the multiplicative update, which uses two edges to update the missing third

edge. The axial attention is also modeled using the bias to the row-wise attention, that is coming from previous pairwise representations. This completes the information flow from pairwise representations to individual molecular neighborhoods.

To improve the training stability of the model we followed the recent observation by Shuangfei et al Zhai *et al.* (2023) that training instability is usually accompanied by the low entropy of the attention layer. We implemented $\sigma$Reparam algorithm to regularize attention weights in all the layers that use attention in the Evoformer block (Appendix algorithm 19). Additionally, following recent efforts by the Ziyao et al Li *et al.* (2022), we replaced all ReLU activation units with Gaussian Error Linear Units (GELU) and added postprocessing layer to the output of the OuterProductMean module.

## 2.6 Losses

We used several loss terms to improve predictions and training stability of the model. To compare predicted structures with the ground truth ones we use frame-aligned point error loss (FAPE) first proposed by the AlphaFold2 team. Given a set of predicted coordinates $x_i, \ i \in 1, \ldots, N$ and reference frames $F_j, \ j \in 1, \ldots, M$, along with a set of true coordinates $x_i^{gt}$ and reference frames $F_j^{gt}$, the loss is computed by transforming coordinates of each point $x_i$ into the frame $F_j$ and comparing them to the corresponding ground truth coordinates $x_i^{gt}$ in the ground truth frames $F_j^{gt}$:

$$FAPE\left(F, x, F^{gt}, x^{gt}\right) = \frac{1}{NM} \sum_{ij} \|F_i \circ x_j - F_i^{gt} \circ x_j^{gt}\|$$

, where $N$ is the number of atoms in a molecule and $M$ is the number of rigid bodies comprising it (see Appendix algorithm 21). The detailed description of the whole algorithm is given in Appendix section A.4.1 and Appendix algorithm 22. The second loss term penalizes the clashes of atoms in the predicted structure. To construct it, we used Van-der-Waals radii for each atom Mantina *et al.* (2009) and assign minimum distance between any two atoms in a molecule as $r_{ij}^{min} = r_i^{VW} + r_j^{VW}$. However, we want to exclude atoms that belong to the same neighborhoods from the loss. Therefore, we set $r_{ij}^{min} = 0$ if atoms $i$ and $j$ are the second-order neighbors in the molecular graph (Appendix algorithm 23). After predicting atomic coordinates for a molecule we calculate the clash penalty as:

$$L_{clash}(x) = \frac{1}{\sum_{ij} r_{ij}^{min} > 0} \sum_{ij} ReLU\left(r_{ij}^{min} - \|x_i - x_j\|\right)$$

It is also worth to note, that some atomic positions are predicted more than once, because of the overlap of rigid neighborhoods along the covalent bonds of a molecule (See Fig. 2). Therefore, we averaged each atomic position across all the rigid neighborhoods containing given atom (Appendix algorithm 20). Finally, during the prediction of rotation matrices (Eq.2) one can face co-linearity problem for the vectors $b_1$ and $b_2$, which are used to parameterize rotation. Hence, we added the co-linearity loss, that minimizes scalar product between the normalized vectors $b_1$ and $b_2$ (Appendix algorithm 24). One of the key features of AlphaFold2 is its ability to predict the quality of the resulting structures, which we aim to retain in our work. For this, we predict the per-neighborhood accuracy of the structure (pLDDT) using a small neural network from the final neighborhood representations (Appendix algorithm 25). All the losses, except for pLDDT are predicted for each iteration of the Evoformer block and then averaged. To take into account isomorphisms of the molecular graph, we compute the minimum FAPE and pLDDT losses over all the isomorphisms.

## 3 Results

The dataset used for training and test is based on PubChemQC Nakata and Shimazaki (2017) consisting of approximately 4 million molecules, where each molecule is represented with Simplified Molecular Input Line Entry Specification (SMILES) description, IUPAC International Chemical Identifier (InChI), and the ground-state and excited-state 3D geometries of these molecules. Here we used the pre-processed version of this dataset, Molecule3D, provided by Xu *et al.* (2021c). The dataset is split into 60% / 20% / 20% subsets for training/validation/test, and two different splitting strategies is used: random and scaffold-based splitting, where a scaffold refers to a molecule's core component consisting of connected rings without branches. Note, that scaffold-based split leads to a distribution shift between the training and test subsets. We used the scaffold-based split, which forces a model to

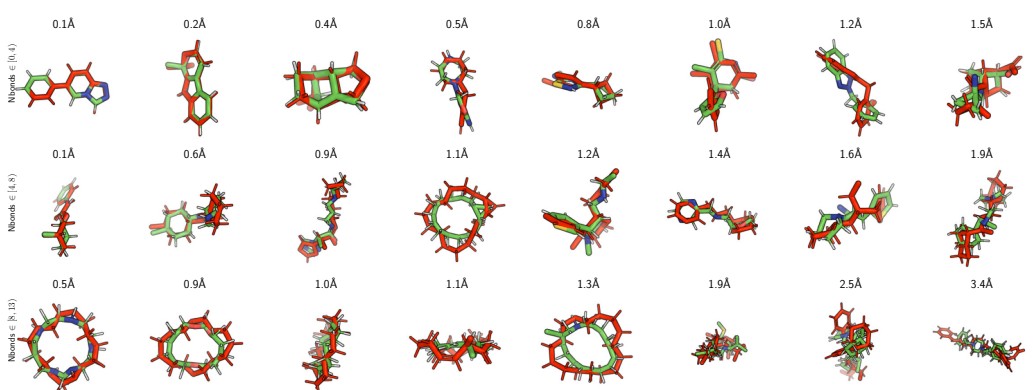

Figure 5: Examples of predicted structures from the test set. Red: true structures; Blue: predicted structures. We split the test set into sets of molecules depending on the number of the rotable bonds and the RMSD of the prediction. Then for each number of rotable bonds, we split RMSD region into 8 equal intervals and picked one example for each.

Table 1: The evaluation of the trained models on the test set of Molecule3D benchmark.

| Model | MAE | RMSD | FAPE | HOMO-LUMO |
|---|---|---|---|---|
| AlphaMol/8 | 0.365 | 0.886 | 1.565 | 0.435 |
| AlphaMol/12 | 0.353 | 0.893 | 1.509 | 0.337 |
| AlphaMol/24 | **0.304** | 0.830 | 1.434 | 0.382 |
| RDKit ETKDG | 0.532 | - | - | **0.1524** |
| DeeperGCN-DAGNN + Distance | 0.660 | - | - | 0.2000 |
| DeeperGCN-DAGNN + Coordinates | 0.763 | - | - | 0.2371 |

capture such distribution shifts in chemical space and measures the out-of-distribution generalization ability of the model. Additionally, we filtered the dataset to exclude molecules containing only one neighborhood, molecules, whose graph has disconnected components, molecules that have pentavalent atoms, and some other cases (see Appendix section A.2). In total, we excluded $\approx 4,000$ structures across the training, validation, and test sets.

We have trained models with different number of Evoformer blocks: small (8), medium(12), and big(24). The model was trained using Adam optimizer with the learning rate $1.5 \cdot 10^{-4}$, without a learning rate schedule. The training of each model was carried out on one node with 4xV100 for $1 \cdot 10^6$ iterations, which approximatelly correspond to 5, 7 and 15 days of node-time for small, medium and big models. To compare our results with the previous algorithms we compute the mean absolute error (MAE) performance metric:

$$MAE = \frac{1}{N^2} \sum_{i,j=1..N, i \neq j} \left| d_{ij}^{pred} - d_{ij}^{data} \right| \tag{5}$$

where $d_{ij}^{pred}$ is the distance between atom $i$ and atom $j$ in the prediction and $d_{ij}^{data}$ is the actual distance between the same atoms.

We compared our models with the DeeperGCN-DAGNN model Liu *et al.* (2021) model, which predicts either distances between atoms or 3D atomic coordinates Xu *et al.* (2021c). Additionally, we used the ETKDG algorithm Riniker and Landrum (2015) implemented in RDKit Landrum (2020) as a baseline. The results are shown in Table 1. One can see, that even the smallest model outperforms previously published methods in predicting 3D structures of ground states of molecules.

The existing methods typically report degradation of the prediction quality with respect to the number of rotatable bonds in a molecule. We did not observed such a drawback for our method, and Figure 6 shows the distribution of RMSD of the predicted structures in the test set versus the number of rotatable bonds. As one can see, the RMSD distribution has the second peak at the RMSD value of $\sim$

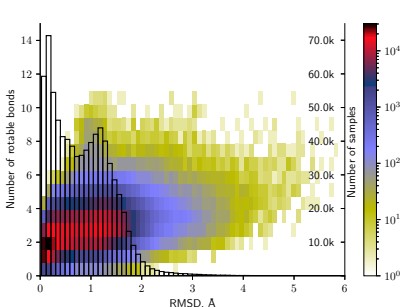 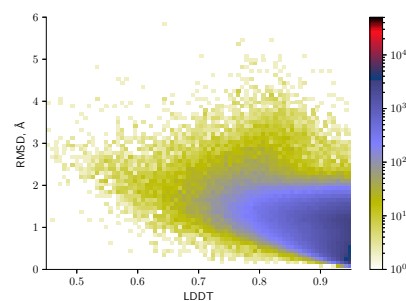

Figure 6: Left axis: distribution of RMSD of predicted structures depending on the number of rotable bonds. Color scheme uses logarithmic scale. Right axis: summary distribution of RMSD scores.

Figure 7: Distribution of RMSD of predicted structures depending on average predicted LDDT of the molecule. Color scheme uses logarithmic scale.

1.3 Åcorresponding to molecules with up to 11 rotatable bonds, while the worst case predictions have about 7 rotatable bonds. Figure 5 shows examples of the predicted molecular structures sampled from three groups with different number of rotatable bonds and 8 groups corresponding to the different RMSD values.

It is important to note, that one of the advantages of our model is ability to estimate the confidence of the predictions, which could be useful in the downstream tasks. Figure 7 shows the correlation between the average predicted LDDT and the RMSD of the predicted structure. Expectedly, we observed a negative correlation between the predicted LDDT and RMSD.

## 4    DISCUSSION

In this study we developed a method to predict ground state small molecule structures based on novel representation of molecular structures as a set of rigid neighborhoods. We introduced how to compute loss functions over all isomorphisms of a molecular structure, as well as how to fix the chirality of the predicted structures. These features are especially relevant for biological molecules; for example, chirality may imply very different bioactivity properties between the small molecule enantiomers and structural properties of protein chains. To the best of our knowledge, the proposed approach of factorizing isomorphisms of the molecules is the only one available that can deal with this long tail molecules. For example, lipids have long tails of carbohydrates for which other approaches to account for isomorphisms fail, because their number grows exponentially with the length of the molecule. To successfully train the model we introduced some changes to the Evoformer block, namely, $\sigma$Reparam methodZhai et al. (2023), scalarization approach to equivarianceDu et al. (2022a) and Graham-Schmidt processZhou et al. (2019) to represent rotations. These changes allowed us to exclude the learning rate scheduling and running exponential averaging and significantly stabilized the training.

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

# A APPENDIX

## A.1 BOND GEOMETRIES

In this work we first remove terminal hydrogens of the molecule using the algorithm 1. We do this in because we want to preserve hydrogens for carbon atoms in SP3 hydridization, while keeping the same number of neighborhoods as with the hydrogen-free molecular representation. In particular we remove all hydrogens bonded to heavy atoms that are connected only to one other heavy atom.

We extract bond geometries and bond distances from the dataset using the algorithm 2.

---

**Algorithm 1** Removing terminal hydrogens

---

**Require:** Mol
  $done \leftarrow False$
  **while** $not\ done$ **do**
    $done \leftarrow True$
    **for** $atom\_i \in Mol.GetAtoms()$ **do**
      **if** $atom\_i.Symbol! =' H'$ **then**
        $Neighb \leftarrow atom\_i.Neighbors()$
        $N_{heavy} = \sum_{atom\_j \in Neighb} \begin{cases} 1, atom\_j.Symbol! =' H' \\ 0, otherwise \end{cases}$   ▷ Number of heavy atoms
connected to $atom\_i$
        $N_{hyd} = len(Neighb) - N_{heavy}$      ▷ Number of hydrogens connected to $atom\_i$
        **if** $N_{heavy} == 1\ \&\&\ N_{hyd} > 0$ **then**      ▷ If $atom\_i$ hydrogens are terminal
          **for** $atom\_j \in Neighb$ **do**
            **if** $atom\_j.Symbol ==' H'$ **then**
              $Mol.RemoveAtom(atom\_j)$
              $done \leftarrow False$
            **end if**
          **end for**
        **end if**
      **end if**
    **end for**
  **end while**
    **return** $Mol$

---

---

**Algorithm 2** An overview of bond geometries extraction

---

**Require:** Dataset
  mol $\in$ Dataset
  **for** mol $\in$ Dataset **do**
    $Hoods \leftarrow \textbf{Neighborhoods}(mol)$
    **for** $hood \in Hoods$ **do**
      $HoodKey \leftarrow Cat(hybridization, num\_single\_bonds, num\_double\_bonds, \dots)$
      $UnitVecs \leftarrow []$
      **for** $(i, j) \in HoodBonds$ **do**     ▷ Index $i$ is always the root atom of the neighborhood
        $BondKey \leftarrow Cat(mol[i].Symbol, mol[j].Symbol)$   ▷ $mol[i].Symbol$ is the atomic
symbol of $i$-th atom
        $BondDist \leftarrow \sqrt{|mol[i].Coords - mol[j].Coords|^2}$
        $\textbf{Store}[BondKey] \leftarrow BondDist$
        $UnitVecs.append\left((mol[j].Coords - mol[i].Coords)/BondDist\right)$
      **end for**
      **if** $len(UnitVecs) >= 3\ \&\&\ \textbf{volume\_sign}(UnitVecs) < 0$ **then**
        $UnitVecs \leftarrow \begin{pmatrix} 1 & 0 & 0 \\ 0 & 1 & 0 \\ 0 & 0 & -1 \end{pmatrix} \cdot UnitVecs$     ▷ Making sure that all geometries have
positive volume sign
      **end if**
      $perms \leftarrow []$
      **for** $perm \in Permutations(range(len(UnitVecs)))$ **do**▷ Permutations of bond indices
        **if** $RMSD(UnitVecs, UnitVecs[perm]) < 0.1$ **then**   ▷ Only counting permutations
for one chirality
          $perms.append(perm)$
        **end if**
      **end for**
      $\textbf{Store}[HoodKey] \leftarrow UnitVecs, perms$
    **end for**
  **end for**

---

Table 2: Previous work on molecular structure prediction

| Method | Year | Inner representation | Equivariance | Training framework |
|---|---|---|---|---|
| GraphDG Simm and Hernández-Lobato (2019) | 2020 | Distances | Scalars | VAE |
| ConfVAE Xu *et al.* (2021a) | 2021 | Distances | Scalars | VAE |
| ConfGF Shi *et al.* (2021) | 2021 | Distances | Scalars | Energy-based |
| DGSM Luo *et al.* (2021) | 2021 | Distances | Scalars | Energy-based |
| CGCF Xu *et al.* (2021b) | 2021 | Distances | Scalars | Energy-based, NeuralODE |
| SDEGen Zhang *et al.* (2023) | 2023 | Distances | Scalars | Diffusion |
| GeoMol Ganea *et al.* (2021) | 2021 | Angles and dist. | Scalars | OT |
| BOKEI Chan *et al.* (2020) | 2020 | Torsions | Scalars | BO |
| Torsional Diffusion Jing *et al.* (2022) | 2022 | Torsions | Scalars | Diffusion |
| CVGAE Mansimov *et al.* (2019) | 2019 | Coordinates | Fixed frame | VAE |
| DMCG Zhu *et al.* (2022) | 2022 | Coordinates | Fixed frame | VAE |
| EVFN Zhang *et al.* (????) | – | Coordinates | Scalarization | Energy-based |
| GeoDiff Xu *et al.* (2022) | 2022 | Coordinates | Scalars | Diffusion |
| EC-Conf Fan *et al.* (2023) | 2023 | Coordinates | Irr. repr. | Diffusion |

Table 3: Previous methods that predict structures of molecules based on molecular graph description. Updated version of the survey Du *et al.* (2022b). We also classify score-matching algorithms as energy-based.

Table 4: Previous work on protein structure prediction

| Method | Inner representation | Equivariance | Training framework |
|---|---|---|---|
| AlphaFold2 Jumper *et al.* (2021) | Rigid bodies | Invariant Point Attention | OT |
| RoseTTAFold Baek *et al.* (2021) | Atomic coordinates | SE(3) Transformer | OT |
| OmegaFold Wu *et al.* (2022) | Rigid bodies | Invariant Point Attention | OT |
| ESMFold Lin *et al.* (2022) | Rigid bodies | Invariant Point Attention | OT |
| SE(3)-Fold Wu *et al.* (2021) | Coordinates | Scalars | Energy-based |
| RFDiffusion Watson *et al.* (2022) | Atomic coordinates | SE(3) Transformer | Diffusion |
| Chroma Ingraham *et al.* (2022) | Rigid bodies | Relative transforms | Correlated Diffusion |

Table 5: Previous methods that predict structures of proteins.

We use algorithm 3 to extract molecular neighborhoods and expression 6 to obtain the volume sign. As we can see we treat a bond geometry as an object that is equal to other geometry up to a permutation of bonds. I.e. only the hybridization of the root atom and the number of single/double/triple bonds counts. We store the fist bond length and bond geometry of a particular class without averaging over all the structures in the dataset. Table 6 shows resulting geometries and the distribution of RMSD values when aligning them to the structures from the dataset. It also contains examples of aligned geometries as well as outliers in terms of alignment RMSD.

$$\begin{cases} \vec{v_i} = \vec{r_i}, & \text{3-neighbor} \\ \vec{v_i} = \vec{r_4} - \vec{r_i}, & \text{4-neighbor} \end{cases}, i \in [1, 2, 3]$$

$$\text{volume\_sign} = \text{sign}(\vec{v_1}, \vec{v_2} \times \vec{v_3})$$

| Signature | Geometry | Initial molecule | Outlier | RMSD distribution |
|---|---|---|---|---|

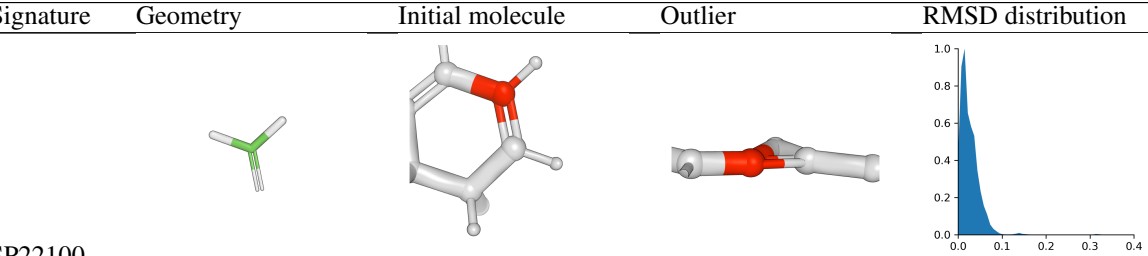

SP22100

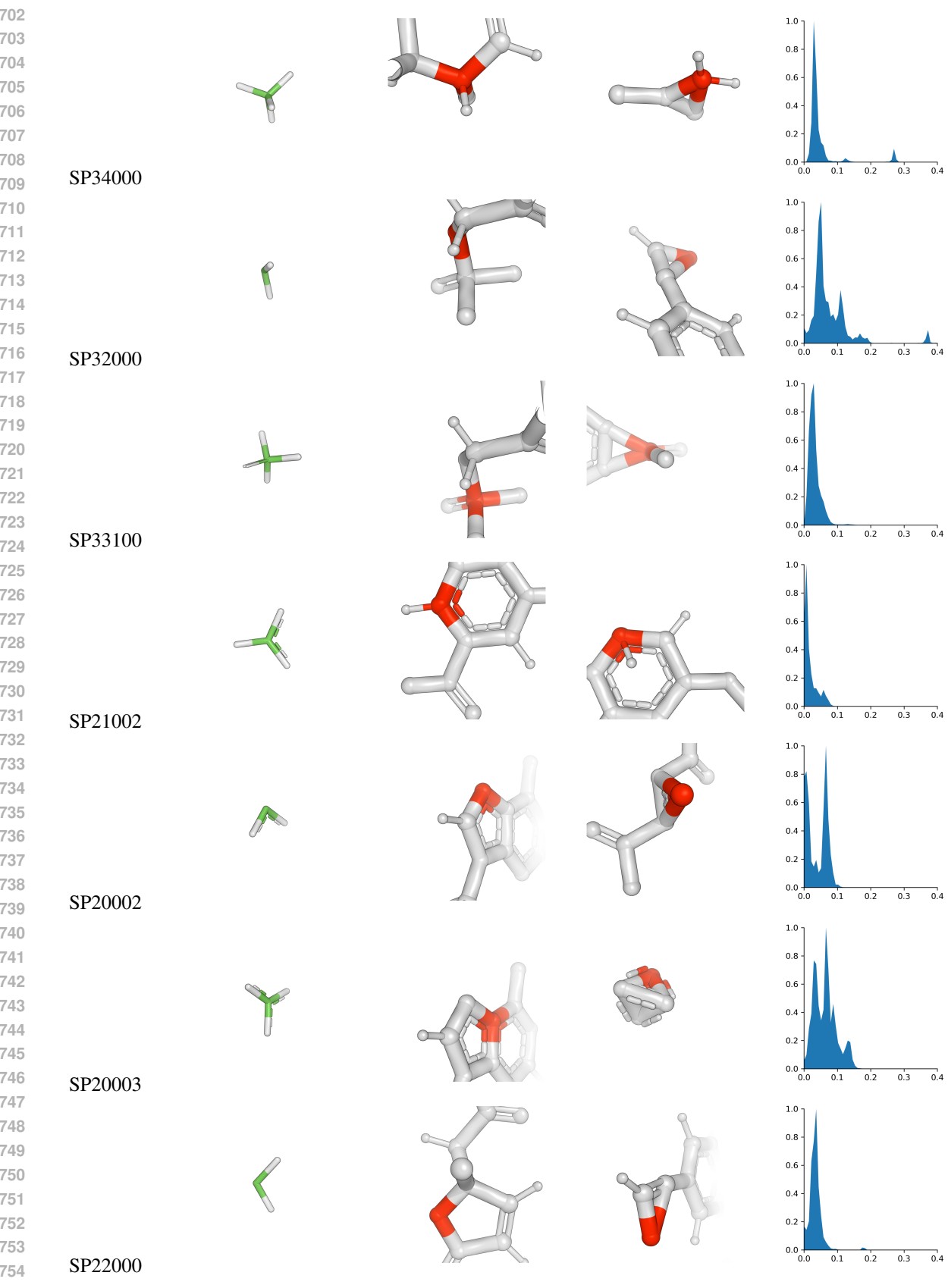

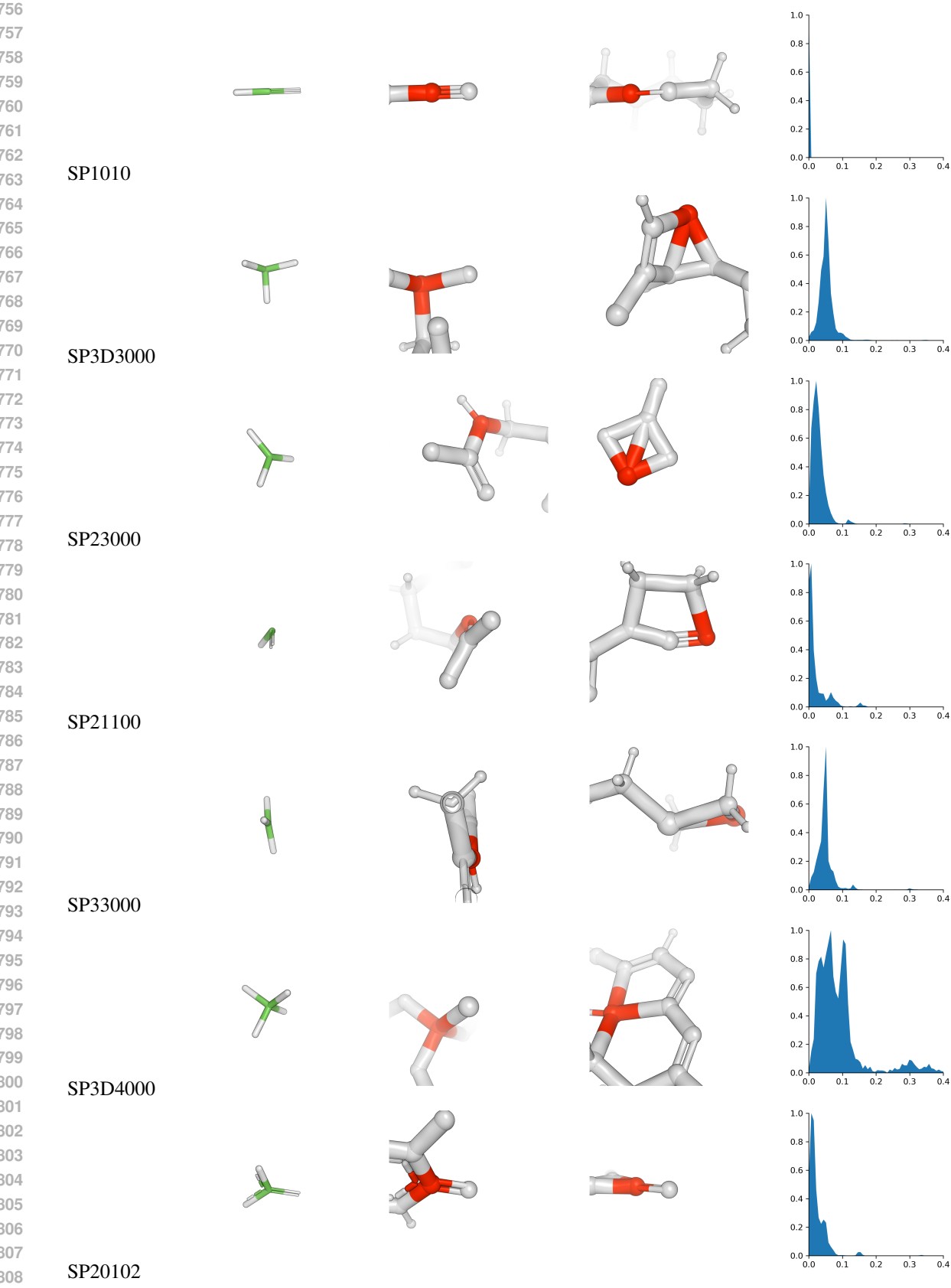

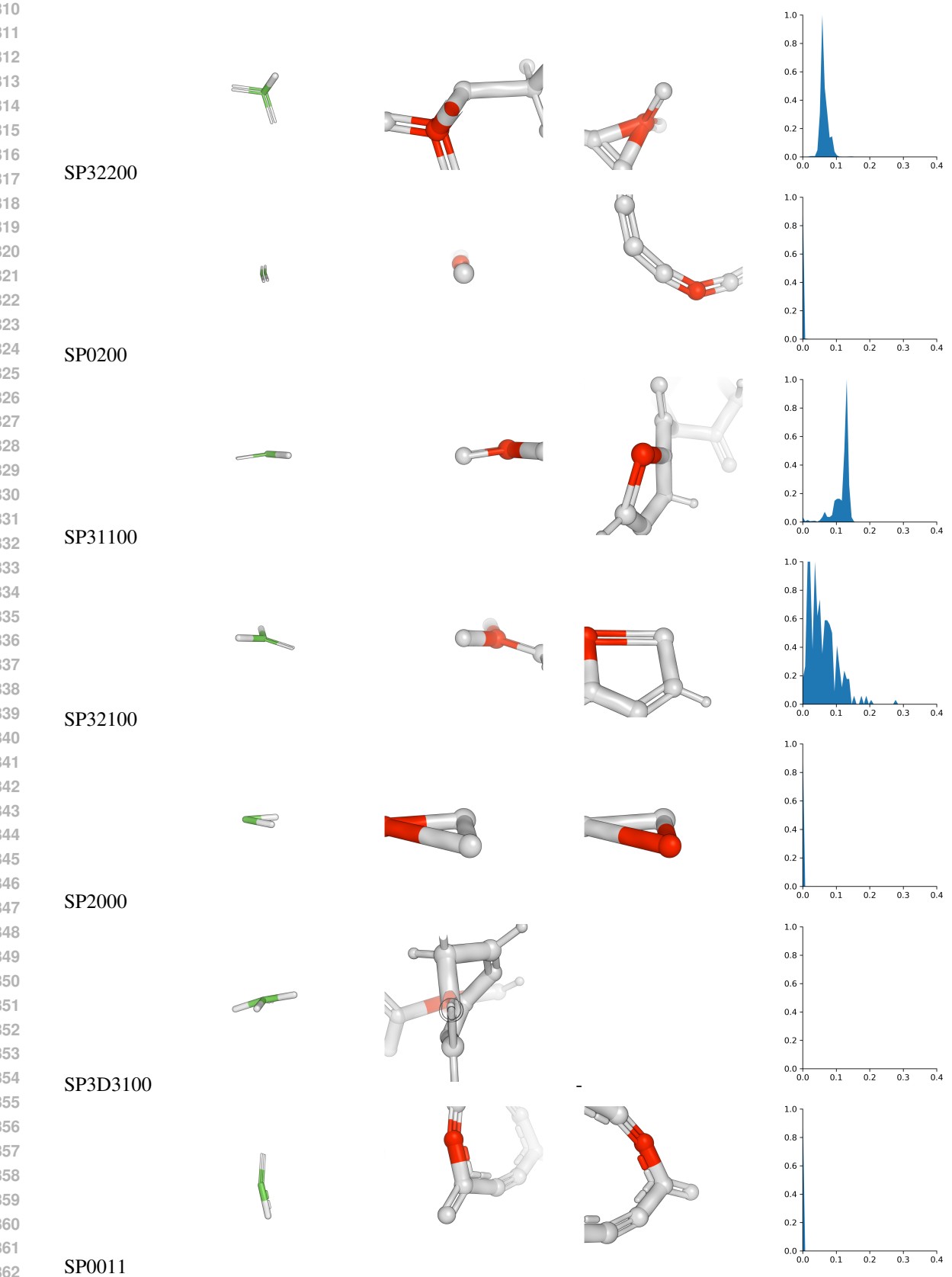

Table 6: Bond geometries.

---

**Algorithm 3** Algorithm for extracting molecular neighborhoods

---

**Require:** mol, rank
  $A_{ij \in [0,N)}, \leftarrow$ **Adjaccency**$(mol)$              ▷ Getting adjacency matrix of the molecule
  $B_{ij \in [0,N)}, \leftarrow$ **BondIndex**$(mol)$              ▷ Bond indices for adjacency matrix
  $HoodIdx \leftarrow 0$
  **for** $k < N$ **do**
    **if** $\sum_i A_{ki} > 1$ **then**                           ▷ Degree of the node k
      $HoodRootAtom[HoodIdx] \leftarrow k$
      $neighbors \leftarrow []$
      **for** $l < N$ **do**
        **if** $A_{kl} == 1$ **then**
          $neighbors.append(l, rank[l])$
        **end if**
      **end for**
      $neighbors \leftarrow$ **sort**$(neighbors, lambda x : x[1])$    ▷ Sorting neighbors according to the
  CIP ranks
      $AtomIdx \leftarrow 0$
      **for** $l \in neighbors$ **do**
        $HoodAtomIndices[HoodIdx, AtomIdx] \leftarrow l$
        $HoodBondIndices[HoodIdx, AtomIdx] \leftarrow B_{kl}$
        $AtomIdx \leftarrow AtomIdx + 1$
      **end for**
      $HoodIdx \leftarrow HoodIdx + 1$
    **end if**
  **end for****return** $HoodRootAtom$, $HoodAtomIndices$, $HoodBondIndices$

---

## A.2 DATA PROCESSING

The data processing pipeline is outlined in algorithms 4 and 5. The pipeline 4 processes molecular graphs without any knowledge of ground truth atomic coordinates. We extract atomic and bond feature sets described in the manuscript first. Then we get the indices of atoms comrising molecular neighborhoods and rearrange per-atom features into per-neighborhood and pairwise features. Afterwards we compute initial coordinates for each neighborhood atoms according to the bonds geometries, that we described in section A.1. Afterwards we use these coordinates to get pairwise transforms between neighborhoods. And finally, we enumerate isomorphisms in the molecule.

The processing of atomic coordinates is outlined in algorithm 5. We generate coordinates for each isomorphisms of the molecular graph and extract transforms between initial neighborhood coordinates and generated ground truth coordinates giving us a set of ground truth transforms.

---

**Algorithm 4** Pipeline of molecular graph pre-processing

---

**Require:** mol                                                     ▷ Molecular graph
  $F_{i \in [0,N)}, \leftarrow$ **AtomicFeatures**$(mol)$              ▷ Getting atomic features
  $B_{j \in [0,M)}, \leftarrow$ **BondFeatures**$(mol)$             ▷ Getting bond features
  $NeighbIdx_{ij} \leftarrow$ **GetNeighborhoods**$(mol)$    ▷ Getting indices of atoms in neighborhoods,
  $i \in [0, M), j \in [0, m_i)]$
  $HoodFeat_{i \in [0,N)} \leftarrow$ **GetHoodFeatures**$(F_i, B_{ij}, NeighbIdx_{ij})$    ▷ Getting neighborhood
  features
  $PairFeat_{ij \in [0,N)} \leftarrow$ **GetPairFeatures**$(mol)$    ▷ Getting pairwise neighborhood features
  $x_{ij}^{init} \leftarrow$ **GetInitCoords**$(mol)$    ▷ Getting initial coordinates of atoms in neighborhoods
  $T_{ij \in [0,N)} \leftarrow$ **GetPairTansforms**$(mol)$    ▷ Getting pairwise transforms of neighborhoods
  $global\_iso, local\_iso \leftarrow$ **GetIsomorphisms**$(mol)$    ▷ Getting isomorphisms of the molecule

---

This pipeline has several special cases shown in the Table 7 along with the number of such cases in the dateset. The first case is the excessive number of isomorphisms: when we try to predict a molecule with hydrogens, these hydrogens can switch places between themselves without changing

---

**Algorithm 5** Pipeline of atomic coordinates pre-processing

---

**Require:** atom_coords, global_iso, local_iso, $x^{init}$, $NeighbIdx$
    **for** $k \in num\_global\_iso$ **do**
        **for** $l \in num\_local\_iso$ **do**
            $iso\_neighbor\_indices_{kl} \leftarrow global\_iso[k] \cdot local\_iso[l] \cdot NeighbIdx$
            $gt\_neighbor\_positions_{kl} \leftarrow atom\_coords[iso\_neighbor\_indices_{kl}]$
        **end for**
    **end for**
    $gt\_neighb\_transforms \leftarrow \textbf{FitNeighbTransforms}(x^{init}, gt\_neighbor\_positions)$

---

the graph. Table 7 show that the number of isomorphism tend to increase exponentially when we have long hydrocarbon tails in some molecules. We describe our solution to this problem in section A.2.2.

Some molecules have several connected components in their graphs. In this case we select the largest one and continue processing. Molecules containing penta-valent silicon compounds($\ldots Si_5 \ldots$) and penta-valent phosphorous ($\ldots CPF_4 \ldots$) are excluded from the dataset. We also exclude molecules contanining only one neighborhood, single atom or atoms rare in biologically releavant compounds (Be, Kr, Ar etc).

| Exception key | Examples | Number |
|---|---|---|
| Isomorphisms |  | 4733 |
| Disconnected |  | 1547 |
| Alignment |  | 21676 |
| Pentavalent |  | 800 |
| Single hood |  | 3841 |

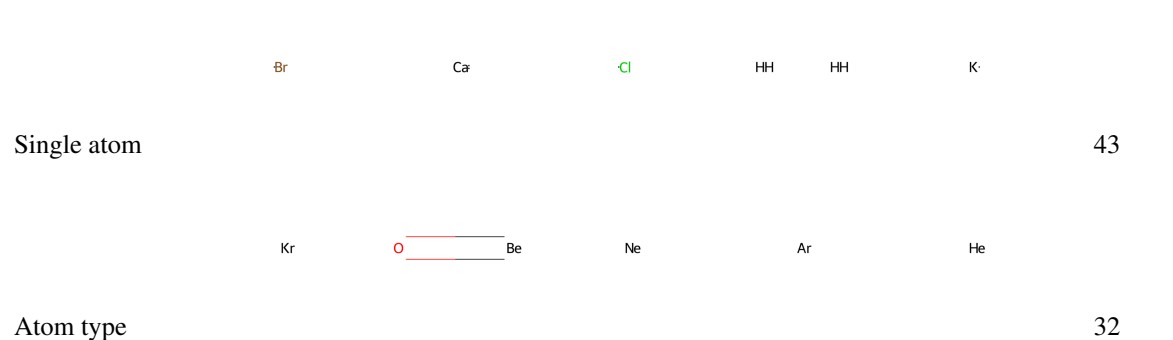

| Single atom | 43 |
| Atom type | 32 |

Table 7: Exclusion set.

### A.2.1 NEIGHBORHOODS

As outlined in the data processing pipeline Alg 4, we first obtain atomic indices for each neighborhood. To make them consistent with chirality of the molecule we sort the indices for each neighborhood according to the CIP ranks of atoms in the molecule (Alg. 6). Then we compute the features of

---

**Algorithm 6 GetNeighborhoods**

---

**Require:** $A_{ij}, \ i, j \in [0, N)$ $\qquad\qquad\qquad\qquad$ ▷ Adjacency matrix of the molecule
**Require:** $CIP_i$ $\qquad\qquad\qquad\qquad\qquad\qquad$ ▷ CIP ranks of atoms in the molecule
$\quad RootIdx = \{i : i \text{ is not leaf node}\}$ $\qquad$ ▷ Every non-leaf atom makes a neighborhood
$\quad$ **for** $i \in RootIdx$ **do**
$\quad\quad NeighbIdx_i \leftarrow \{j : A_{ij} = 1\}$
$\quad\quad$ **Sort**$(NeighbIdx_i, key = CIP_i)$ $\qquad$ ▷ Sorting indexes using CIP ranks of the atoms
$\quad$ **end for**
$\quad$ **return** $NeighbIdx_i$

---

the neighborhoods following Alg 7. The atomic and bond features ($F$ and $B$) are described in the manuscipt. Pairwise features for neighborhoods are computed using algorithm 8, that is similar to the

---

**Algorithm 7 GetHoodFeatures**

---

**Require:** $edge_j, \ j \in [0, M)$ $\qquad\qquad\qquad$ ▷ Indexes of atoms connected by the M bonds
**Require:** $F_i, \ i \in [0, N)$ $\qquad\qquad\qquad\qquad\qquad\qquad\qquad$ ▷ Atomic features
**Require:** $B_j, \ j \in [0, M)$ $\qquad\qquad\qquad\qquad\qquad\qquad\qquad$ ▷ Bond features
**Require:** $RootIdx, NeighbIdx$ $\qquad\quad$ ▷ Root and neighbor indices for each neighborhood
$\quad$ **for** $i \in RootIdx$ **do**
$\quad\quad$ **for** $j \in NeighbIdx_i$ **do**
$\quad\quad\quad bond\_index \leftarrow \{k : edge_k = (i, j)\}$
$\quad\quad\quad HoodFeat_{ij} \leftarrow \mathbf{cat}(F_i, F_j, B_{bond\_index})$
$\quad\quad$ **end for**
$\quad$ **end for**
$\quad$ **return** $HoodFeat_{ij}$

---

previous algorithm.

Additionaly, we have to provide geometric input features to our model. We compute the initial geometry of a neighborhood by combining bond geometry and bond lengths, extracted from the data and described in the section A.1. Algorithm 9 outlines the procedure we use to assign initial coordinates to the atoms of neighborhoods. We first assign signatures to the neighborhoods in the same way we did it in section A.1, then we load bond order, allowed permutations and bond lengths from the data geometric data. We then find permutation that matches the current bond order to the one in the data and assing the coordinates according to this permutation.

**Algorithm 8 GetPairFeatures**

**Require:** $edge_j, \ j \in [0, M)$            ▷ Indexes of atoms connected by the M bonds
**Require:** $F_i, \ i \in [0, N)$                        ▷ Atomic features
**Require:** $B_j, \ j \in [0, M)$                       ▷ Bond features
**Require:** $RootIdx$                ▷ Root indices for each neighborhood
  **for** $i \in RootIdx$ **do**
    **for** $j \in RootIdx$ **do**
      $bond\_index \leftarrow \{k : edge_k = (i, j)\}$
      **if** $bond\_index \neq \emptyset$ **then**
        $PairFeat_{ij} \leftarrow \mathbf{cat}(F_i, F_j, B_{bond\_index})$
      **end if**
    **end for**
  **end for**
  **return** $PairFeat_{ij}$

**Algorithm 9 GetInitCoords**

**Require:** $edge_j, \ j \in [0, M)$        ▷ Indexes of atoms connected by the M bonds
**Require:** $BondType_j, \ j \in [0, M)$              ▷ Bond types
**Require:** $Permutations_i, \ i \in [0, N)$      ▷ Allowed permutations for each neighborhood
**Require:** $BondOrder_i, \ i \in [0, N)$         ▷ Bond order for each neighborhood
**Require:** $Coords_{ij}, \ i \in [0, N), \ j \in [0, K_i)$    ▷ Coordinates of unit vectors of geometry of each neighborhood
**Require:** $BondLength_{ij}, \ i \in [0, N), \ j \in [0, K_i)$ ▷ Bond lengths of each bond in neighborhoods
**Require:** $RootIdx, NeighbIdx$         ▷ Root and neighbor indices for each neighborhood
  **for** $i \in RootIdx$ **do**
    $bond\_order \leftarrow \{\}$
    **for** $j \in NeighbIdx_i$ **do**         ▷ Extracting bond order of the current neighborhood
      $bond\_index \leftarrow \{k : edge_k = (i, j)\}$
      $bond\_order \leftarrow bond\_order \cup \{BondType_{bond\_index}\}$
    **end for**
    $perm \leftarrow \{p : p(bond\_order) = BondOrder_i, p \in Permutations_i\}$    ▷ Matching the bond orders
    $init\_coords_i \leftarrow \{(0, 0, 0)\}$         ▷ Placing the root atom in the center
    **for** $k \in perm(NeighbIdx_i)$ **do**         ▷ Computing the initial coordinates
      $init\_coords_{ik} \leftarrow Coords_{ik} * BondLength_{ik}$
    **end for**
  **end for**
  **return** $init\_coords_{ik}$

Table 8: Each one-hot feature is an encoding of a given property + 1 bit that indicates that the property is abnormal or incorrectly assigned.

| Feature | Encryption | Size |
|---|---|---|
| Atom type | one-hot | 35 |
| Is aromatic | 1/0 | 1 |
| Atom degree | one-hot | 8 |
| Atom hybridization | one-hot | 6 |
| Atom implicit valence | one-hot | 8 |
| Atom formal charge | one-hot | 4 |
| Size of the ring atom belongs to | one-hot | 6 |
| Number of rings atom belongs to | one-hot | 5 |
| Atom chirality to | 1/0 | 1 |
| Bond type | one-hot | 4 |
| Bond in ring | 1/0 | 1 |
| Bond is conjugated | 1/0 | 1 |
| Bond is aromatic | 1/0 | 1 |
| Bond chirality | 1/0 | 1 |

Finally we obtain pair transforms using algorithm 10. For each pair of neighborhoods $i$ and $j$ that are connected by a bond we take the coordinates of atom in the first neighborhood $vec\_src$ that constitutes the bond $ij$. Similarily we take the coordinates of atom from the second neighborhood $j$ $vec\_dst$ that belongs to the bond $ji$ between neighborhoods. Then we compute the transform that aligns second neighborhood along the bond $vec\_src$.

---

**Algorithm 10 GetPairTansforms**

---

**Require:** $edge_j, \ j \in [0, M)$ ▷ Indexes of atoms connected by the M bonds
**Require:** $init\_coords_{ik}, \ i \in [0, N), \ k \in [0, K_i)$ ▷ Initial coordinates of neighborhood atoms
**Require:** $RootIdx, NeighbIdx$ ▷ Root and neighbor indices for each neighborhood
  **for** $i \in RootIdx$ **do**
    **for** $j \in RootIdx$ **do**
      **if** $\{k : edge_k = (i, j)\} \neq \emptyset$ **then**
        $vec\_src \leftarrow \{init\_coords_{ik} : NeighbIdx_{ik} = j\}$ ▷ Initial coordinates of atom from neighborhood $i$ to $j$
        $vec\_dst \leftarrow \{init\_coords_{jk} : NeighbIdx_{jk} = i\}$ ▷ Initial coordinates of atom from neighborhood $j$ to $i$
        $H_{src} \leftarrow AlignX(vec\_src)$ ▷ Alignment matrix of vector to the X-axis
        $H_{dst} \leftarrow AlignX(vec\_dst)$
        $T_{ij}^{rot} \leftarrow H_{src}^T \cdot H_{flip} \cdot H_{dst}$ ▷ Rotational part of the pair transform
        $T_{ij}^{trans} \leftarrow vec\_src$ ▷ Translational part of the pair transform
      **end if**
    **end for**
  **end for**
  **return** $T_{ij}$

---

The final set of features of each neighborhood is given in the Table 8.

A.2.2 ISOMORPHISMS FACTORIZATION

Lets denote a set of all isomorphisms of a molecule graph as $I$. Any one element in the set $I$ is a permutation of atomic indices of a molecule. We need to enumerate this set in order to compute the final loss of the model prediction:

$$L = \min_{iso \in I} FAPE\left(T_{pred}, X_{pred}, T_{gt}(iso), X_{gt}(iso)\right) \tag{6}$$

where $T_{pred}, T_{gt}$ are the predicted and ground truth transforms of the neighborhoods and $X_{pred}, X_{gt}$ are the predicted and ground truth atomic coordinates of the molecule. The Eq.6 represents the

form we use in the current work, alternatively we can take the minimum over $T_{pred}(iso^{-1})$ and $X_{pred}(iso^{-1})$. However this alternative formulation makes algorithm more computationally heavy. The $X_{gt}(iso)$ can be written as $X_{gt}(iso) = \{x_{gt}^{iso(i)}, \ i \in [0, N)\}$. $T_{gt}(iso)$ are the transforms from initial coordinates of the neighborhood atoms to the ground truth coordinates permutted using an elemenet of the set $I$.

Naively we can enumerate all isomorphisms of a molecule graph by first coloring the graph vertexes using the atom type and graph edges using the bond type. However, this procedure will yield the number of isomorphisms that exceed $10^5$ for all molecules in the first row of Table 7. The reason for this is that each hydrogen bonded to a carbon using signle bond can be swapped for any other hydrogen of the same atom, so that the number of such swap combinations grows exponentially with the number of carbons in the molecule.

In this work we deal with this problem for the case of terminal hydrogen atoms only. To circumvent combinatorial explosion of the isomorphism set, we factorize the set into local and global isomorphisms. Local isomorphisms $I_i^{(l)}$ are computed for each neighborhood $i$. Each local isomorphism should only permute hydrogens, leaving heavy atoms in their respective places. Global isomorphisms $I^{(g)}$ are the isomorphisms of the molecule without hydrogens, that are extended to the added hydrogens with identity permutation.

**Proposition:** $\forall i \in I \exists l \in I^{(l)}, m \in I^{(g)} : i = l \cdot m$ We leave out the proof of this proposition, however it should be trivial. If the proposition holds, we compute the loss of the model prediciton in the following way:

$$iso_i^{(l)} = \underset{iso \in I_i^{(l)}}{\text{argmin}} \ FAPE\left(T_{pred}, X_{pred}, T_{gt}(iso), X_{gt}(iso)\right), \ i \in [0, M) \quad (7)$$

$$L = \min_{iso \in I^{(g)}} FAPE\left(T_{pred}, X_{pred}, T_{gt}\left(iso \times \prod_{i \in [0,M)} iso_i^{(l)}\right), X_{gt}\left(iso \times \prod_{i \in [0,M)} iso_i\right)\right) \quad (8)$$

Despite this method, there are still some ligands that have highly symmetric molecular graphs. Although the number of global isomorphisms they have is on the order of $10^4$, we still exclude 91 molecules from the dataset that have more than 512 global isomorphisms for convenience. Figure 8 shows some examples of such molecules.

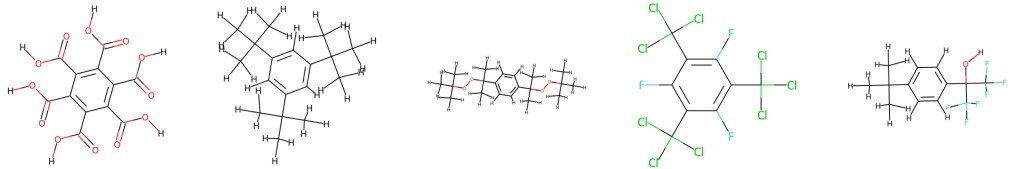

Figure 8: Examples of molecules with the graph that has more than 512 global isomorphisms.

### A.2.3 GROUND TRUTH

While obtaining neighborhood transforms from neighbor positions we run into two special cases: the chirality of a neighborhood is not set and the corresponding bond geometry fits the coordinates badly ($RMSD - \min RMSD > 0.3$). The algorithm 11 outlines the way we detect and treat these special cases. If the $chirality\_mask$ is true, then we assign chirality to this atom and reprocess the data. In practice, we observed that quatro-valent phosphorus in certain lingands is not labeled as chiral. If the $fit\_mask$ contains true values, we leave transforms that we obtained in this algorithm and unmask all its local isomorphisms.

### A.2.4 DATASET STATISTICS

Overall out of 3,982,254 molecules in the dataset we exclude 5105. Out of which are 801 compounds containing penta-valent atoms, 4137 single-neighborhood compounds, 43 examples that have atoms

**Algorithm 11 FitNeighbTransforms**

---

**Require:** $init\_coords_{ik}, \ i \in [0, N), \ k \in [0, K_i)$ ▷ Initial coordinates of atoms in neighborhoods
**Require:** $gt\_positions_{glik}, \ g \in global\_iso, \ l \in local\_iso, \ i \in [0, N), \ k \in [0, K_i)$ ▷ Ground truth coordinates
$rmsd_{gli} \leftarrow RMSD(init\_coords, gt\_positions)$ ▷ Aligning all initial coordinates to all ground truth ones
$min\_rmsd_i \leftarrow \min\limits_{gi}(rmsd_{gli})$ ▷ Min RMSD for each neighborhood
$iso\_mask_{gli} \leftarrow (rmsd_{gli} - min\_rmsd_i) < 0.3$ ▷ Flagging true all variants that are close to the best fit
$loc\_iso\_mask_{gi} \leftarrow \exists_l iso\_mask_{gli}$
**if** $\forall_g \neg(\forall_i loc\_iso_m ask_{gi})$ **then**
    ▷ If all global isomorphisms have at least one hood that does not fit any local isomorphisms
$$W_{gi} \leftarrow \begin{pmatrix} 1 & 0 & 0 \\ 0 & 1 & 0 \\ 0 & 0 & 2loc\_iso\_mask_{gi} - 1 \end{pmatrix}$$
                        ▷ We flip chirality of neighborhoods that did not fit any local isomorphism
    $rmsd_{gli}^{chir} \leftarrow RMSD(W \cdot init\_coords, gt\_positions)$
    $loc\_iso\_mask_{gi}^{chir} \leftarrow \exists_l rmsd_{gli}^{chir} < 0.82$
                        ▷ If fit changes when we flip the chirality then it is assigned incorrectly
    $chirality\_mask \leftarrow (\exists_g loc\_iso\_mask_{gi}) \oplus (\exists_g loc\_iso\_mask_{gi}^{chir})$
    **if** $\exists_i chirality\_mask_i$ **then**
        Flip chiral center $i$ and redo data processing
    **end if**
                        ▷ If the fit does not change, then we have non-standard neighborhood geometry
    $fit\_mask \leftarrow (\neg \exists_g loc\_iso\_mask_{gi}) \wedge (\neg \exists_g loc\_iso\_mask_{gi}^{chir})$
    **if** $\exists_i fit\_mask_i$ **then**
        $iso\_mask[fit\_mask] \leftarrow True$
    **end if**
**end if**
            ▷ If there is no local isomorphism for at least one neighborhood, we mask such global isomorphism
$glob\_iso\_mask_g \leftarrow \forall_i \exists_l iso\_mask_{gli}$
$iso\_mask[\neg glob\_iso\_mask_g] \leftarrow False$
$loc\_iso\_mask_{gi} \leftarrow \exists_l iso\_mask_{gli}$
**if** $\forall_g \neg(\forall_i loc\_iso_m ask_{gi})$ **then**
    ▷ If after all the changes we still have problematic neighborhoods we retain those close to the optimal one
    $loc\_iso\_min\_idx_{gi} = \operatorname*{argmin}\limits_l(rmsd_{gli})$
    $opt\_rmsd_g \leftarrow \sum_i \min\limits_l(rmsd_{gli})$ ▷ Min RMSD for each isomorphism
    $glob\_iso\_mask_g \leftarrow opt\_rmsd_g - \min\limits_g(opt\_rmsd_g) < 0.3$
    $iso\_mask_{gli} \leftarrow False$
    $iso\_mask_{gloc\_iso\_min\_idx_{gi}i} \leftarrow True$
    $iso\_mask_{g \neg glob\_iso\_mask_g} \leftarrow False$
**end if**
                        ▷ Finally we obtain rotations and translations from alignment matrix $U$
**return** $rot \leftarrow rmsd_{gli}.U^T$
**return** $trans \leftarrow gt\_positions_{gl0k}$
**return** $iso\_mask$

---

without neighbors, 32 compounds with rare atom types and 1 compound that is disconnected and has non-trivial chirality. Additionally we remove 91 examples because of excessive number of global isomorphisms.

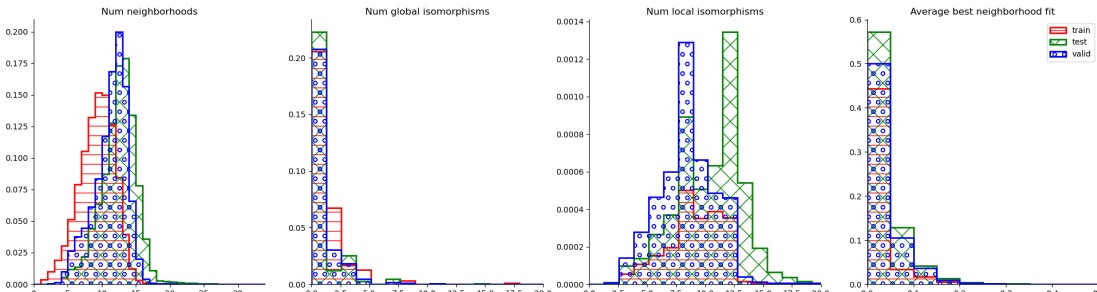

Figure 9: Distibution of number of neighborhoods, global and local isomorphisms and average neighborhood best fit rmsd in training, validation and test sets.

Figure 9 shows the distribution of the number of neighborhoods in train, test and validation subsets, as well as distribution of number of global and local isomorphisms and average neighborhood rmsd fit. We see that test set is slightly more challenging than the training and validation subsets. We also can see in Fig. 9 that the overal best fit of the dataset using our molecule representation is good enough to assume the validity of such a representation.

Additionally, Figure 10 shows molecules that are outliers in the statistics shown on the Fig 9. The first row shows largest molecules in the dataset with the labels corresponding to the numbers of neighborhoods in each molecule. The second row corresponds to the molecules with the biggest numbers of global isomorphisms. The third row shows molecules with the biggest number of neighborhoods with non-trivial local isomorphisms (that are not filtered out based on the structure). Finally, the fourth row shows the molecules that have the worst fit using rigid-body bond geometries along with the average rmsd over all neighborhoods.

### A.3 MODEL

Algorithm 12 gives and overwiev of the model. First, we generate initial transforms of the neighborhoods and embed input features. Then we iterate over Evoformer blocks, and iteratively refine initial translations and rotations. Iteration parameters (Linear layers and Evoformer paramteres) are unique. During the iteration we first scalarize the geometric features, pass them as a transformer input to the Evoformer and then vectorize the transformed Evoformer output, which we use to update the geometry of a molecule.

In this work we follow the approach by the AlphaFold2 team and use the same initialation of the rigid body transforms. Algorithm 13 shows that we place all the neighborhoods in the frame origin and assign them the same rotation.

For feature embedding we take the output features of the data processing algorithm $HoodFeat$ and $PairFeat$. First we append initial atomic coordinates of the neighbors in neighborhoods and linearly transform them, obtaining neighbor embeddings $neighb\_embed$ as a result. Next we concatenate neighbor embedding for each neighborhood and transform these features obtaining embeddings for neighborhoods $hood\_embed$. The pairwise neighborhood features are passed through a linear transform followed by ReLU and another linear transform to obtain pairwise embeddings $pair\_embed$. Algorithm 14 shows the pseudocode of the whole process.

To make evoformer application on the geometric features equivariant we first centralize the coordinates of neighborhoods (Algorithm 15). This gives us translation invariance. The rotation equivariance is archived by transforming geometric features into scalars in certain basis and after applying Evoformer, transforming the output back into geometric features using the same basis. To do this we first construct a set of basis for a molecule, one for each neighborhood. We use rotation of the neighborhood as a vector basis (Algorithm 16, $node\_basis$). To scalarize rotation matrixes we also construct the basis

**Algorithm 12 AlphaMol**

---

**Require:** $HoodFeat, PairFeat, init\_coords$
  $rot, trans \leftarrow InitialTransforms()$
  $single\_act, pair\_act \leftarrow FeatureEmbedding(HoodFeat, PairFeat, init\_coords)$
  $all\_rot, all\_trans \leftarrow \emptyset$     $\triangleright$ Ordered (only the last position order matters) sets of rotations and translations
  **for** $i \in [0, num\_evoformer\_blocks)$ **do**
                                        $\triangleright$ Scalarization of the geometry
    $trans \leftarrow Centralize(trans)$
    $node\_basis, edge\_basis \leftarrow GetBasis(rot)$
    $node\_scalars, edge\_scalars \leftarrow Scalarize(rot, trans, pair\_rot, pair\_trans, node\_basis, edge\_basis)$
                                        $\triangleright$ Evoformer part
    $single\_act \leftarrow Linear(single\_act \bigoplus node\_scalars)$
    $pair\_act \leftarrow Linear(pair\_act \bigoplus edge\_scalars)$
    $single\_act, pair\_act \leftarrow Evoformer(single\_act, pair\_act)$
    $transform \leftarrow Linear(single\_act)$
                              $\triangleright$ Vectorization and geometry update
    $new\_rot, new\_trans \leftarrow Vectorize(transform, node\_basis, rot, trans)$
    $rot \leftarrow rot \cdot new\_rot$
    $all\_rot \leftarrow all\_rot \cup rot$                             $\triangleright$ Saving new rotation
    $trans \leftarrow trans + new\_trans$
    $all\_trans \leftarrow all\_trans \cup trans$                     $\triangleright$ Saving new translation
  **end for**
  **return** $all\_rot, \ all\_trans, \ single\_act, \ pair\_act$

---

**Algorithm 13 InitialTransforms**

---

                                          $\triangleright$ Using the same initialization as AlphaFold2
**Require:** $batch\_size, num\_hoods$
  $rot \leftarrow \begin{pmatrix} 1 & 0 & 0 \\ 0 & 1 & 0 \\ 0 & 0 & 1 \end{pmatrix}.repeat(batch\_size, num\_hoods)$

  $trans \leftarrow \begin{pmatrix} 0 \\ 0 \\ 0 \end{pmatrix}.repeat(batch\_size, num\_hoods)$

  **return** $rot_{bijk}, trans_{bij}, b \in [0, batch\_size), i \in [0, num\_hoods), j, k \in [0, 3)$

---

**Algorithm 14 FeatureEmbedding**

---

**Require:** $HoodFeat_{bij} \in R^{N_{feat}}, init\_coords_{bij} \in R^3, i \in [0, N_{hoods}), \ j \in [0, N_{neighbors})$
**Require:** $PairFeat_{bij} \in R^{M_{feat}}, i, j \in [0, N_{hoods})$
  $HoodFeat_{bij} \in R^{N_{feat}+3} \leftarrow HoodFeat_{bij} \bigoplus init\_coords_{bij}$
  $neighb\_embed_{bij} \leftarrow relu(Linear(relu(Linear(HoodFeat_{bij}))))$
  $neighb\_embed_{bi} \in R^{N_{neighbors}*N_{feat}} \leftarrow \bigoplus_j neighb\_embed_{bij}$
  $hood\_embed_{bi} \leftarrow Linear(relu(Linear(neighb\_embed_{bi})))$
  $pair\_embed_{bij} \leftarrow Linear(relu(Linear(PairFeat_{bij})))$
  **return** $hood\_embed_{bi}, pair\_embed_{bij}$

---

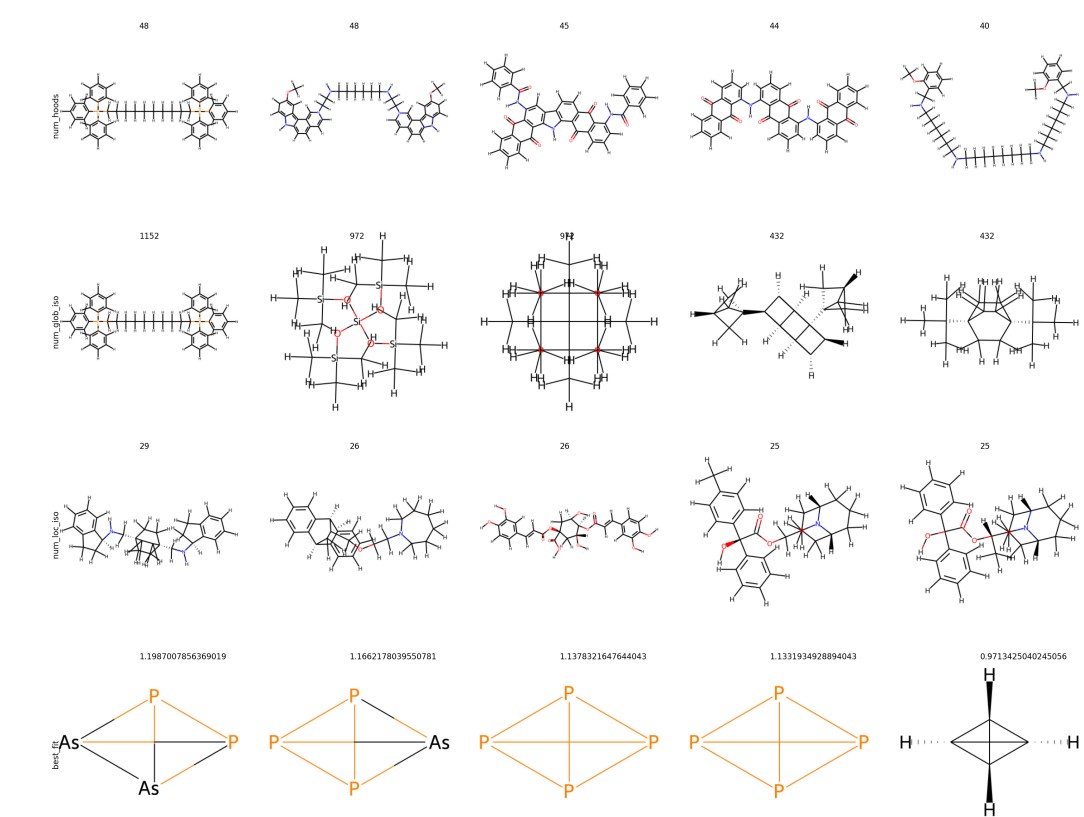

Figure 10: Outliers from the datasets. Labels on the left denote the category, labels above denote the parameter, that this category has. The values of paramteres correspond to the data on Fig 9.

in the space of (1,1)-tensors. We do it by taking outer product of vector basis of each node with all the other nodes. This way we obtain $N_{hoods}^2$ basis for pairwise tensor features (Algorithm 16, $edge\_basis$).

---

**Algorithm 15 Centralize**

---

**Require:** $trans_{bi} \in R^3, \ i \in [0, N_{hoods})$
$\quad center_b \in R^3 \leftarrow \frac{1}{N_{hoods}} \sum_i trans_{bi}$
$\quad$ **return** $trans_{bi} - center_b$

---

**Algorithm 16 GetBasis**

---

**Require:** $rot_{bi} \in R^9, \ i \in [0, N_{hoods})$
$\quad node\_basis_{bik} \in R^3 \leftarrow rot_{bi}^T, \ k \in [0, 3)$
$\quad edge\_basis_{bijkl} \in R^9 \leftarrow node\_basis_{bik} \bigotimes node\_basis_{bjl}, \ k, l \in [0, 3)$
$\quad$ **return** $node\_basis_{bik}, edge\_basis_{bijkl}$

---

To obtain scalar features from translations, rotations of each neighborhood and the pairwise translations and rotations features we first project translations onto vectors of the $node\_basis$ (Algorithm 17) obtaining $node\_scalars$. For the pairwise translations we first compute relative translations $rel\_trans$ and then transform them into translation tensor by taking outer product of their components (Algorithm 17, $edge\_trans$). Then we project the relative translations tensor onto the pairwise tensor basis. Similarily we obtain scalar pairwise rotations. Finally we concatenate pairwise scalars into $edge\_scalars$.

---

**Algorithm 17 Scalarize**

---

**Require:** $trans_{bi} \in R^3, rot_{bi} \in R^9 \quad i \in [0, N_{hoods})$
**Require:** $pair\_trans_{bij} \in R^3, pair\_rot_{bij} \in R^9 \quad i, j \in [0, N_{hoods})$
**Require:** $node\_basis_{bim} \in R^3, edge\_basis_{bijkl} \in R^9 \quad i, j \in [0, N_{hoods}), \quad m \in [0, 3), \quad k, l \in [0, 9)$
    $node\_scalars_{bi} \in R^3 \leftarrow (trans_{bi}, node\_basis_{bim})$
    $rel\_trans_{bij} \in R^3 \leftarrow (trans_{bi} - trans_{bj}) + pair\_trans_{bij}$
    $edge\_trans_{bij} \in R^9 \leftarrow rel\_trans_{bij} \bigotimes rel\_trans_{bij}$
    $scalar\_edge\_trans_{bij} \in R^9 \leftarrow (rel\_trans_{bij}, edge\_basis_{bijkl})$
    $scalar\_edge\_rot_{bij} \in R^9 \leftarrow (pair\_rot_{bij}, edge\_basis_{bijkl})$
    $edge\_scalars_{bij} \in R^{18} \leftarrow edge\_trans_{bij} \bigoplus scalar\_edge\_rot_{bij}$
    **return** $node\_scalars_{bi} \in R^3, edge\_scalars_{bij} \in R^{18}$

---

Algorith 18 describes our vectorization process of the output features $transform_{bim}$. Here we obtain three vectors for each neighborhood by treating $transform_{bim}$ as the decomposition coefficients into the $node\_basis_{bi}$. One of these three vectors is then interpreted as a translation update of a neighborhood $translation_{bi}$. The other two vectors are used to construct rotation using Gramm-Schmidt process.

---

**Algorithm 18 Vectorize**

---

**Require:** $transform_{bim} \in R^3, \quad i \in [0, N_{hoods}), \quad m \in [0, 3)$
**Require:** $node\_basis_{bim} \in R^3, edge\_basis_{bijkl} \in R^9 \quad i, j \in [0, N_{hoods}), \quad m \in [0, 3), \quad k, l \in [0, 9)$
    $vectors_{bik} \in R^3 \leftarrow \sum_m (node\_basis_{bik}, transform_{bim})$
    $translation_{bi} \in R^3 \leftarrow vectors_{bi0}$
    $a_{bi}^{(1)} \in R^3 \leftarrow vectors_{bi1}$
    $a_{bi}^{(2)} \in R^3 \leftarrow vectors_{bi2}$
                                             ▷ Gramm-schmidt process
    $b_{bi}^{(1)} \leftarrow \frac{a_{bi}^{(1)}}{|a_{bi}^{(1)}|}$
    $b_{bi}^{(2)} \leftarrow a_{bi}^{(2)} - (b_{bi}^{(1)}, a_{bi}^{(2)}) b_{bi}^{(1)}$
    $b_{bi}^{(2)} \leftarrow \frac{b_{bi}^{(2)}}{|b_{bi}^{(2)}|}$
    $b_{bi}^{(3)} \leftarrow b_{bi}^{(1)} \times b_{bi}^{(2)}$
    $rotation_{bi} \in R^9 \leftarrow b_{bi}^{(1)} \bigoplus b_{bi}^{(2)} \bigoplus b_{bi}^{(3)}$
    **return** $translation_{bi} \in R^3, rotation_{bi} \in R^9$

---

### A.3.1 EVOFORMER

In this work we use the same Evoformer block, as the one in AlphaFold2 with few modifications. First, we do not need column-wise attention, because the input single features have only one row. The second important change is that throughout the Evoformer block we use GELU activation function instead of ReLU. The major change we made is the addition of spectral normalization in the attention layer. Algorithm 19 shows the changes to the gated self-attention with pair bias in bold. Similar changes are done to the triangle attention modules.

### A.4 LOSSES

First we compute the iterative atomic structures of a molecule based on rotations and translations ($all\_rot, \ all\_trans$) output of the model. Algorithm 20 shows the outline of molecule reconstruction using the scatter operation. Effectively we predict atom positions belonging to the bond between neighborhoods twice and then average over these predictions. We ommit the technical details of tensor manipulation. Similar procedure is performed for the single representations of the neighborhoods. In

---

**Algorithm 19 Gated self-attention with pair bias**

---

**Require:** $m_{bi} \in R^{N_{feat}}, \quad i \in [0, N_{hoods})$
**Require:** $z_{bij} \in R^{N_{feat}}, \quad i \in [0, N_{hoods})$
$\qquad\qquad\qquad\qquad\qquad\qquad\qquad\qquad\qquad$ ▷ Iteratively compute maximum eigenvalue of the matrix $K^T Q$
$\quad \mathbf{W} \leftarrow \mathbf{K^T Q}$
$\quad \mathbf{u} \leftarrow \mathbf{W} \cdot \mathbf{u}$ $\qquad\qquad\qquad\qquad\qquad$ ▷ $u$ is the parameter of this module, saved for the next step
$\quad \mathbf{u} \leftarrow \frac{\mathbf{u}}{|\mathbf{u}|}$
$\quad \mathbf{v} \leftarrow \mathbf{W} \cdot \mathbf{v}$ $\qquad\qquad\qquad\qquad\qquad$ ▷ $v$ is the parameter of this module, saved for the next step
$\quad \mathbf{v} \leftarrow \frac{\mathbf{v}}{|\mathbf{v}|}$
$\quad \sigma^{\mathbf{h}} \leftarrow \sum_{\mathbf{dc}} \mathbf{u_d^h W_{dc}^h v_c^h}$
$\qquad\qquad\qquad\qquad\qquad\qquad\qquad$ ▷ Standard gated self-attention with pair bias
$\quad m_{bi} \leftarrow LayerNorm(m_{bi})$
$\quad q_{bi}^h, k_{bi}^h, v_{bi}^h \leftarrow LinearNoBias^{QKV}(m_{bi})$ $\qquad$ ▷ Q, K, V are matrixes of the linear transform
$\quad b_{bij}^h \leftarrow LinearNoBias(LayerNorm(z_{bij}))$
$\quad g_{bi}^h \leftarrow Sigmoid(Linear(m_{bi}))$
$\quad a_{bij}^h \leftarrow Softmax_j(\frac{\mathbf{1}}{\sigma^{\mathbf{h}}\sqrt{\mathbf{c}}} q_{bi}^{h\,T} k_{bj}^h + b_{bij}^h)$ $\qquad\qquad$ ▷ Additional factor $\sigma^h$
$\quad o_{bi}^h \leftarrow g_{bi}^h \cdot \sum_j a_{bij}^h v_{bi}^h$
$\quad \tilde{m}_{bi} \leftarrow Linear(concat_h(o_{bi}^h))$
$\quad$ **return** $\tilde{m}$

---

the end we have $atom\_positions_{lbk}$ tensor, where $l$ indexes evoformer blocks outputs, $b$ corresponds to the molecule index in a batch and $k$ enumerates the atoms in a molecule. Additionally we obtain $single\_act_{bk}$ representation for each atom in the batch of molecules.

---

**Algorithm 20 Structure reconstruction**

---

**Require:** $all\_rot_{lbi} \in R^9, all\_trans_{lbi} \in R^3, \quad l \in [0, num\_evoformer\_blocks), \quad b \in [0, batch\_size), \quad i \in [0, N_{hoods})$
**Require:** $init\_coords_{bik} \in R^3, \quad i \in [0, N_{hoods}), \quad k \in [0, N_{neighb})$
**Require:** $atom\_mask_{bik}, \quad i \in [0, N_{hoods}), \quad k \in [0, N_{neighb})$ ▷ 1/0 depending on whether atom $k$ is present in neighborhood $i$
**Require:** $atom\_indices_{bik}, \quad i \in [0, N_{hoods}), \quad k \in [0, N_{neighb})$ $\qquad$ ▷ global index of atom $k$ in neighborhood $i$
$\quad neighbor\_positions_{lbik} \leftarrow all\_rot_{lbi} \cdot init\_coords_{bik} + all\_trans_{lbi}$
$\quad num\_atoms_m \leftarrow \sum_{bik} atom\_mask_{bik}\delta(atom\_indices_{bik} - m)$ $\qquad$ ▷ Scatter operation: we sum over $atom\_mask$ to the cell with indices of $atom\_index$
$\quad atom\_positions_{lm} \leftarrow \frac{1}{num\_atoms_m} \sum_{bik} neighbor\_positions_{lbik}\delta(atom\_indices_{bik} - m)$
$\quad atom\_positions_{lbk} \in R^3 \leftarrow atom\_positions_{lm}, \quad k \in [0, num\_atoms_b)$ $\qquad$ ▷ Rearranging tensor to the batch of molecules
$\quad single\_act_m \leftarrow \frac{1}{num\_atoms_m} \sum_{bik} single\_act_{bik}\delta(atom\_indices_{bik} - l)$
$\quad single\_act_{bk} \in R^{N_{feat}} \leftarrow single\_act_m, \quad k \in [0, num\_atoms_b)$
$\quad$ **return** $atom\_positions_{lbk}, single\_act_{bk}, neighbor\_positions_{lbik}$

---

### A.4.1 STRUCTURAL LOSSES

The first loss that we compute is Frame-Aligned Point Error(FAPE, Algorithm 21) of the reconstructed structure with respect to the ground truth structure. However, because we have our ground truth data in the factorized form we first have to find the best matching global and local isomorphism. Algorithm 22 outlines our implementation. The key feature of this algorithm is that it is an approximation of the Eq.7. We do not reconstruct structures for each isomorphism, instead we compute FAPE for neighborhoods for each isomorphism and then approximate FAPE of the whole structure by the sum FAPE over all neighborhoods. In practice this approximation gives us the same minimum as the whole-structure FAPE. Afterwards we reconstruct ground truth structure for the selected isomorphisms and compute the correct FAPE score for the whole reconstructed structure.

1512
1513
1514
1515

---

**Algorithm 21 Frame-Aligned Point Error(FAPE)**

---

**Require:** $pred\_T_i \in R^{12}, pred\_pos_j \in R^3, gt\_T_i \in R^{12}, gt\_pos_j \in R^3$

1516  $x_{ij} \leftarrow pred\_T_i^{-1} \circ pred\_pos_j$

1517  $gt\_x_{ij} \leftarrow gt\_T_i^{-1} \circ gt\_pos_j$

1518  $d_{ij} = \sqrt{||x_{ij} - gt\_x_{ij}||^2}$

1519  **return** $\frac{1}{10 N_i N_j} \sum_{ij} (\min(10, d_{ij}))$

---

1521
1522
1523
1524
1525
1526
1527

---

**Algorithm 22 Structure loss**

---

**Require:** $gt\_neighbor\_positions_{bglin} \in R^3, gt\_rot_{bgli} \in R^9, gt\_trans_{gli} \in R^3, \quad b \in [0, batch\_size), \ g \in [0, N_{glob\_iso}), \ l \in [0, N_{loc\_iso_i}) \ i \in [0, N_{hoods})$

**Require:** $all\_rot_{mbi} \in R^9, all\_trans_{mbi} \in R^3, atom\_positions_{mbk} \in R^3 \quad m \in [0, num\_evoformer\_blocks)$

**Require:** $neighbor\_positions_{mbin}$

**Require:** $atom\_mask_{bin}, \ i \in [0, N_{hoods}), \ n \in [0, N_{neighb})$ ▷ 1/0 depending on whether atom $n$ is present in neighborhood $i$

**Require:** $atom\_indices_{bin}, \ i \in [0, N_{hoods}), \ n \in [0, N_{neighb})$ ▷ global index of atom $n$ in neighborhood $i$

▷ Computing FAPE loss for each neighborhood and isomorphism between the prediction and the ground truth

$pred\_rigids_{mbi} \in R^{12} \leftarrow all\_rot_{mbi} \oplus all\_trans_{mbi}$

$gt\_rigids_{bgli} \in R^{12} \leftarrow gt\_rot_{bgli} \oplus gt\_trans_{bgli}$

$neighb\_fape_{mbgli} \leftarrow FAPE(pred\_rigids_{mbi}, gt\_rigids_{bgli}, neighbor\_positions_{mbin}, gt\_neighbor\_positions_{bglin})$

▷ Getting indices of local and global isomorphisms

$local\_iso\_idx_{mbgi}, \ min\_local\_iso_{mbgi} \leftarrow \underset{l}{\operatorname{argmin}}(neighb\_fape_{mbgli}), \ \underset{l}{\min}(neighb\_fape_{mbgli})$

$global\_iso\_idx_{mb} \leftarrow \underset{g}{\operatorname{argmin}}(\sum_i min\_local\_iso_{mbgi})$

$local\_iso\_idx_{mbi} \leftarrow local\_iso\_idx_{m,b,global\_iso\_idx_{mb},i}$

▷ Selecting rigid transforms and neighbor positions of the ground truth based on local and global isomorphism

$gt\_rigids_{mbi} \leftarrow gt\_rigids_{b,global\_iso\_idx_{mb},local\_iso\_idx_{mbi},i}$

$gt\_neighbor\_positions_{mbin} \leftarrow gt\_neighbor\_positions_{b,global\_iso\_idx_{mb},local\_iso\_idx_{mbi},i,n}$

▷ Reconstructing atomic positions for each molecule from neighbor positions, same operation as in Structure reconstruction algorithm

$num\_atoms_f \leftarrow \sum_{bin} atom\_mask_{bin} \delta(atom\_indices_{bin} - f)$

$gt_a tom\_positions_{mf} \leftarrow \frac{1}{num\_atoms_f} \sum_{bin} gt\_neighbor\_positions_{mbin} \delta(atom\_indices_{bin} - f)$

$gt\_atom\_positions_{mbk} \in R^3 \leftarrow gt\_atom\_positions_{mf}, \ k \in [0, num\_atoms_b)$ ▷ Rearranging tensor to the batch of molecules

▷ Computing FAPE loss for the reconstructed ground truth for each evoformer block and molecule in the batch

$struct\_fape_{mb} \leftarrow FAPE(pred\_rigids_{mbi}, gt\_rigids_{mbi}, atom\_positions_{mbk}, gt\_atom\_positions_{mbk})$

**return** $\frac{\sum_{mb} struct\_fape_{mb}}{num\_evoformer\_blocks \cdot batch\_size}, \ gt\_atom\_positions_{mbk}$

---

Additionally we penalize the clashes between atoms in the structure. Algorithm 23 shows our procedure for computing clash loss. Importantly, we exclude first and second neighbors from the loss, because it is guranteed that some of these neighbors belong to the same neighborhoods and our algorithm treats them as rigid bodies. To compute second-order neighbors we use well known formula $A^{second} = A^{first}(A^{first})^T > 0$, where $A^{first}$ is the adjacency matrix of the molecular graph.

---

**Algorithm 23 Clash loss**

---

**Require:** $atom\_positions_{bk} \in R^3, \ b \in [0, batch\_size), \ k \in [0, N_{atoms})$

**Require:** $adj_{bkl}, \ l \in [0, N_{atoms})$ $\qquad\qquad\qquad\qquad$ ▷ Adjacency matrix

**Require:** $r_{bk}, \ k \in [0, N_{atoms})$ $\qquad\qquad\qquad\qquad\qquad$ ▷ Atomic radius

$\quad second\_adj_{bkl} \leftarrow (\sum_m adj_{bkm} adj_{blm}) > 0$ $\qquad$ ▷ Adjacency for second neighbors

$\quad min\_dist_{bkl} \leftarrow (r_{bk} + r_{bl})(1 - second\_adj_{bkl})$

$\quad d_{bkl} \leftarrow \sqrt{||atom\_positions_{bk} - atom\_positions_{bl}||^2}$

$\quad L = \frac{1}{\sum_{bkl}(min\_dist_{bkl} > 0)} \sum_{bkl} ReLU(min\_dist_{bkl} - d_{bkl})$

$\quad$**return** $L$

---

Finally, collinearity of the predictions is also penalized as described in the Algorithm 24. Specifically we save collinearity values during the vectorization stage described by the Algorithm 18. After computing all the Evoformer iterations, we average over the batch and iterations and obtain the loss.

---

**Algorithm 24 Collinearity loss**

---

**Require:** $b_{mbi}^{(1)} \in R^3, a_{mbi}^{(2)} \in R^3 \ m \in [0, num\_evoformer\_blocks), \ b \in [0, batch\_size), \ k \in [0, N_{hoods})$

$\quad coll_{mbi} \leftarrow (b_{mbi}^{(1)}, \frac{a_{mbi}^{(2)}}{||a_{mbi}^{(2)}||})$

$\quad L \leftarrow \frac{1}{num\_evoformer\_blocks \cdot batch\_size \cdot N_{hoods}} \sum_{mbi} coll_{mbi}$

$\quad$**return** $L$

---

Similar to AlphaFold2 we predict the model confidence over its own structure prediction. In our case we predict per-neighborhood lDDT scores. As the Algorithm 25 shows, our implementation has almost no changes from the one used in AlphaFold2.

---

**Algorithm 25 pLDDT loss**

---

**Require:** $atom\_positions_{bk}, gt\_atom\_positions_{bk}, single\_act_{bi}, \ b \in [0, batch\_size), \ i \in [0, N_{hoods}), \ k \in [0, N_{atoms})$

**Require:** $neighbor\_atom\_indices_{bin}, \ n \in [0, N_{neighb})$

**Require:** $v^{(bins)} \in R^{N_{bins}}$ $\qquad\qquad\qquad$ ▷ Vector of bin cutoff values, f.e $[1, 3, 5, \ldots 99]$

$\quad$ ▷ Computing ground truth LDDT, based on the predicted and gt atomic structures, then averaging atomic lDDT over neighborhoods

$\quad d_{bkl} \leftarrow \sqrt{||atom\_positions_{bk} - atom\_positions_{bl}||^2}$

$\quad gt\_d_{bkl} \leftarrow \sqrt{||gt\_atom\_positions_{bk} - gt\_atom\_positions_{bl}||^2}$

$\quad L1_{bkl} \leftarrow |d_{bkl} - gt\_d_{bkl}|$

$\quad score_{bkl} \leftarrow \frac{1}{4}\left((L1_{bkl} < 0.5) + (L1_{bkl} < 1.0) + (L1_{bkl} < 2.0) + (L1_{bkl} < 4.0)\right)$

$\quad gt\_LDDT_{bk} \leftarrow \frac{1}{\sum_l (gt\_d_{bkl} < 15)} \sum_l score_{bkl}(gt\_d_{bkl} < 15)$

$\quad gt\_LDDT_{bi} \leftarrow \frac{1}{N_{neighb}} \sum_n gt\_LDDT_{b, neighbor\_atom\_indices_{bin}}$

$\quad$ ▷ Computing predicted LDDT and using cross-entropy loss to compare it to groud-truth LDDT

$\quad y_{bi} \in R^{N_{bins}} \leftarrow relu(Linear(relu(Linear(LayerNorm(single\_act_{bi})))))$

$\quad p_{bi} \leftarrow SoftMax(Linear(y_{bi}))$

$\quad gt\_p_{bi} \leftarrow OneHot(gt\_LDDT_{bi}, v^{(bins)})$

$\quad pLDDT_{bi} \leftarrow p_{bi}^T v^{(bins)}$

$\quad L \leftarrow \frac{1}{batch\_size \cdot N_{hoods}} \sum_{ib} \left(gt\_p_{bi}^T \log p_{bi}\right)$

$\quad$**return** $pLDDT_{bi}, L$

---

### A.4.2 PROPERTY PREDICTION

We added property prediction head to check whether the atom-wise single activations can be used in the end-to-end fashion for predictions. We use standard SchNet architecture with the assumption of a fully-connected atomic graph. Algorithms 26, 27 and 28 summarize the architecture of the neural network used for HOMO-LUMO gap prediction as well as the loss computation.

---

**Algorithm 26 Property prediction head**

---

**Require:** $atom\_positions_{bk} \in R^3$, $single\_act_{bk}$, $b \in [0, batch\_size)$, $k \in [0, N_{atoms})$
**Require:** $gt_b$          ▷ Ground-truth homo-lumo gap
  $h_{bk} \leftarrow LayerNorm(Linear(single\_act_{bk}))$
  $d_{bkl} \leftarrow \sqrt{||atom\_positions_{bk} - atom\_positions_{bl}||^2}$
  $rbf_{bkl} \leftarrow RBF(d_{bkl})$
  **for** $i \in [0, N_{interact})$ **do**
    $h_{bk} \leftarrow h_{bk} + Interaction_i(h_{bk}, rbf_{bkl})$
  **end for**
  $pred_b \leftarrow \sum_k Linear\left(ShiftedSoftPlus(Linear(h_{bk}))\right)$
  $L \leftarrow \frac{1}{batch\_size} \sum_b |pred_b - gt_b|$
  **return** $L$

---

**Algorithm 27 Radial basis function**

---

**Require:** $d_{bkl}$, $b \in [0, batch\_size)$, $k, l \in [0, N_{atoms})$
**Require:** $x_0 = 0.0$, $x_1 = 5.0$, $N_{gaussians} = 50$
  $r_m \leftarrow x_0 + \frac{m}{N_{gaussians}}(x_1 - x_0)$, $m \in [0, N_{gaussians})$
  $dec_{bklm} \leftarrow e^{-\frac{(d_{bkl} - r_m)^2}{2(r_1 - r_0)^2}}$
  **return** $dec_{bkl} \in R^{N_{gaussians}}$

---

**Algorithm 28 Interaction block**

---

**Require:** $h_{bk} \in R^{N_{feat}}$, $rbf_{bkl} \in R^{N_{gaussians}}$, $b \in [0, batch\_size)$, $k, l \in [0, N_{atoms})$
  $h \leftarrow Linear(h)$
         ▷ CF convolution module
  $W_{bkl} \in R^{N_{feat}} \leftarrow ShiftedSoftPlus(Linear(ShiftedSoftPlus(Linear(rbf_{bkl}))))$
  $h_{bk} \leftarrow \sum_l (h_{bk} \cdot W_{bkl})$
         ▷ Output of interaction block
  $h_{bk} \leftarrow Linear(Softplus(Linear(h_{bk})))$
  **return** $h_{bk} \in R^{N_{feat}}$

---

