# OpenReview forum: "AlphaMol: rigid neighborhood representation for small molecule structure prediction"
_ICLR.cc/2025/Conference — ICLR 2025 Conference Withdrawn Submission_

### Official Review · Reviewer_Xmpf · 2024-10-22

**Soundness:** 2
**Presentation:** 1
**Contribution:** 2
**Rating:** 3
**Confidence:** 5

**Summary:**

This work proposes a method to predict ground state small molecule structures based on novel representation of molecular structures as a set of rigid neighborhoods, which manages to tackle the equivariance and chirality issues during the prediction.

**Strengths:**

1. Inspired by AF2, this work leverages the concept of rigid body and pairwise representations for small molecules and provides a way to encode bonds between rigid bodies.
2. This work specifically addresses issues related to chirality and equivariance in structural prediction, aligning with the intrinsic properties of molecular systems.

**Weaknesses:**

1. The writing of the paper is messy, including the lack of emphasis on key points and paragraph divisions, some formulas without serial numbers, incorrect in-paragraph citation format, etc.
2. Though AlphaMol innovates in molecular representation, it completely stacks the architecture of EvoFormer to form its own network and does not compare with general molecular representation by ablation experiments, thus the conclusion may not seem convincing.
3. The experiments are insufficient since no other stronger baselines are chosen for fair comparison, for instance, DGSM [1], DMCG [2], GeoDiff [3], etc. In addition, the more recognized dataset for evaluation in the field of small molecule structure prediction should be the Geom benchmark [4].

**Questions:**

1. Please reorganize the article with necessary paragraph divisions, use the correct in-paragraph citation format and provide serial numbers of all formulas.
2. AF2 treats residues as rigid bodies, which can simplify protein representation. However, small molecules that are much smaller than proteins can be represented at the atomic level with little overhead. Please further give the reason for using rigid body representation in this work.
3. Could you provide further details on the method of splitting the molecule into different rigid bodies? I also wonder about the size of the vocabulary of all different rigid bodies.
4. Please explain why the results of the baselines on RMSD and FAPE were not reported in Table 1.
5. For fair comparison, it is encouraged to add stronger baselines like DGSM, DMCG, GeoDiff, etc. Moreover, please show me the reason for using Molecule3D rather than the more frequently used benchmark Geom.

**Reference**

[1] Luo, S., Shi, C., Xu, M., & Tang, J. (2021). Predicting molecular conformation via dynamic graph score matching. Advances in Neural Information Processing Systems, 34, 19784-19795.

[2] Zhu, J., Xia, Y., Liu, C., Wu, L., Xie, S., Wang, Y., ... & Liu, T. Y. (2022). Direct molecular conformation generation. arXiv preprint arXiv:2202.01356.

[3] Xu, M., Yu, L., Song, Y., Shi, C., Ermon, S., & Tang, J. (2022). Geodiff: A geometric diffusion model for molecular conformation generation. arXiv preprint arXiv:2203.02923.

[4] Axelrod, S., & Gomez-Bombarelli, R. (2022). GEOM, energy-annotated molecular conformations for property prediction and molecular generation. Scientific Data, 9(1), 185.

**Details Of Ethics Concerns:**

No concerns.

---

### Official Review · Reviewer_Ki62 · 2024-11-04

**Soundness:** 3
**Presentation:** 3
**Contribution:** 3
**Rating:** 8
**Confidence:** 3

**Summary:**

This study introduces a novel approach for predicting ground-state structures of small molecules. The author used a distinctive representation based on rigid neighborhoods. The proposed method computes loss functions over all molecular isomorphisms and addresses chirality constraints—an important consideration in biological molecules, where structural differences between enantiomers can have substantial impacts on bioactivity. A key innovation is the factorization of molecular isomorphisms, which allows the model to handle complex, long-tail molecules like lipids with extended carbohydrate chains, an area where other methods often fall short. The authors also enhanced training stability by modifying the Evoformer block, incorporating the Reparam method, a scalarization approach for equivariance, and the Gram-Schmidt process to manage rotations, which removed the need for learning rate scheduling and running exponential averaging.

**Strengths:**

The study’s unique approach to molecular structure representation via rigid neighborhoods and isomorphism factorization is innovative and well-suited for handling complex structures in biological molecules. By explicitly incorporating chirality, the model addresses crucial aspects of molecular bioactivity, making it particularly valuable for biological and pharmaceutical applications. Additionally, the adjustments to the Evoformer block significantly improve model stability, underscoring the technical soundness of this work.

**Weaknesses:**

The evaluation could be expanded to strengthen the model’s applicability. Currently, the study only benchmarks the AlphaMol model on Molecular 3D tasks, and while various tasks in structure generation are summarized, these benchmarks do not include comparisons to other established models. Evaluating the new model against benchmark models, such as graph-based methods like DimeNet [1], would provide a more comprehensive assessment of its strengths. Including a wider array of molecular representation tasks, such as property prediction or drug-drug interaction prediction, in recent molecular representation surveys [2], would also enhance the study’s impact.

[1] Gasteiger, et al. "Directional message passing on molecular graphs via synthetic coordinates." Advances in Neural Information Processing Systems 34 (2021): 15421-15433.
[2] Guo, Zhichun, et al. "Graph-based molecular representation learning." arXiv preprint arXiv:2207.04869 (2022).

**Questions:**

Could the model’s performance be further validated by comparing it to other benchmark models, especially graph-based methods like DimeNet, across a variety of molecular representation tasks?
How would the model perform on additional tasks that assess molecular representations, such as property prediction or drug-drug interaction prediction, which could offer a broader understanding of its capabilities?
Are there any specific challenges or computational constraints when applying this model to large or highly complex compound libraries?
How does the model handle the challenge of molecular conformation ensembles, which are critical for accurately capturing the dynamic and flexible nature of molecules in real environments? Since conformation ensembles are increasingly recognized as essential for describing realistic molecular behavior [1], further explanation on this aspect could strengthen the model’s relevance, particularly in applications requiring a comprehensive representation of molecular flexibility and bioactivity.

[1] Zhu, Yanqiao, et al. "Learning Over Molecular Conformer Ensembles: Datasets and Benchmarks." The Twelfth International Conference on Learning Representations. 2023.

---

### Official Review · Reviewer_SwGk · 2024-11-04

**Soundness:** 2
**Presentation:** 2
**Contribution:** 2
**Rating:** 3
**Confidence:** 2

**Summary:**

The authors propose a method called AlphaMol, which represents molecular structures as rigid bodies to predict small molecule ground-state structures. AlphaMol defines a rigid body as a set of atoms connected by covalent bonds. The structural information of the rigid body is predefined, and AlphaMol predicts the orientations of the rigid bodies to reconstruct the 3D structure of the complete molecule. The method was evaluated on the Molecule3D dataset and showed comparable performance with other competitive methods.

**Strengths:**

The idea of representing molecular structures as rigid bodies is novel. The authors also provided an in-depth overview of relevant methods for molecular structure generation and a detailed explanation of the AlphaMol method.

**Weaknesses:**

- For Table 1, are the results averaged over multiple runs? Including the standard deviation or confidence interval would improve clarity. Why is the RMSD value missing for RDKit and other methods? Since structure prediction is a long-standing task, incorporating additional methods for comparison would be beneficial.
- In Appendix Table 6, the molecule visualization is incomplete. Some parts of the molecule are missing, making it difficult to interpret the results.
- Figure 7: Quantitatively measuring the correlation between RMSD and LDDT using Spearman's correlation or Kendall's tau might provide better insight than a figure alone.
- Given the multiple possible pairings between two adjacent rigid bodies, how computationally efficient is the proposed method? Can you provide runtime comparisons to existing methods or discuss the computational complexity of their approach relative to the number of rigid bodies or molecule size?

**Questions:**

- I appreciate the detailed overview of relevant methods for structure generation. Since this is also a long-standing task in the scientific community, it might be useful to comment on (or compare with) the performance of widely used tools such as Auto3D (https://auto3d.readthedocs.io/en/latest/index.html), CREST (https://crest-lab.github.io/crest-docs/), OpenBabel (https://open-babel.readthedocs.io/en/latest/index.html), or Orca (https://www.faccts.de/docs/orca/5.0/tutorials/). Some of these tools are computationally expensive, so it's fine to compare on a small subset of molecules.
- The paper might benefit from clearer formula statements and naming conventions. For example, definitions for several symbols are missing from lines 247 to 260.
- It might be clearer for readers if Figure 6 were split into two separate figures.

---

### Official Review · Reviewer_yZ5c · 2024-11-04

**Soundness:** 3
**Presentation:** 3
**Contribution:** 3
**Rating:** 5
**Confidence:** 2

**Summary:**

This paper attempts to extend the methods of AlphaFold2 to small molecule systems, making two main modifications: 1. Constructing rigid body representations; 2. Modifying the Evoformer block. Additionally, it addresses the issue of improving the chirality of structures by factorizing isomorphisms of the molecules.

**Strengths:**

The article is very clear in its writing and organization. It introduces a representation based on rigid bodies, which demonstrates a certain level of originality. The design addressing the chirality of structures provides a reference for future work.

**Weaknesses:**

Although the writing structure of the article is clear, its focus is on the construction of rigid representation, which is not clearly described in the main text, making it somewhat difficult to read. The experimental results in the article are overly simplistic; for example, improvements were made to the Evoformer block, but detailed comparative results were not provided. The explanation of the results is not clear enough; for instance, Table 1 shows the HOMO-LUMO values but does not provide an explanation. Overall, the extension of AlphaFold2 is not a very innovative idea.

**Questions:**

1. The rigid body used is related to the molecular structures in the dataset. Will this method perform consistently on other datasets? How will it handle new molecular structures that are not within the coverage? What hyperparameters can be adjusted in the division of the rigid body? How is the reasonableness of this assumption ensured?

2. What are the effects of the two improvements made to the Evoformer block? Are there any ablation studies conducted?

3.Figure 1 did not clearly explains the entire process, and Figure 4 did not highlight the most important representation part of this article.

4. Will the effect of using rigid body representation be the same as using a pooling layer along with the chirality correction?

5. When comparing RDKit ETKDG, DeeperGCN-DAGNN, and AlphaMol, was the impact of model complexity taken into account?

---

> ### Comment · Reviewer_yZ5c · 2024-11-26
> **Do you have any response？**
>
> as the title

---

### Official Review · Reviewer_teKc · 2024-11-04

**Soundness:** 2
**Presentation:** 1
**Contribution:** 2
**Rating:** 3
**Confidence:** 4

**Summary:**

The primary objective of the paper is to establish a robust architecture for modeling diverse molecular types, ultimately aiming to create a generalized framework capable of managing various molecular modalities. The authors address the small molecule conformer generation challenge by conceptualizing it as a connected set of rigid bodies. They extract commonly available rigid bodies that serve as fundamental building blocks for the majority of molecules in their dataset. A  featurization scheme is introduced, incorporating aspects of chirality and symmetry, with these features subsequently processed through modified sequential Evoformer blocks to produce 2D molecular geometries.

**Strengths:**

- The conceptual framing of small molecule conformations as sets of rigid bodies is interesting and offers a new perspective on molecular modeling.
- The integration of a featurization scheme that considers chirality and symmetry enhances the model's relevance in accurately representing molecular structures.

**Weaknesses:**

- **Clarity Issues**:
    - The clarity of the manuscript is notably poor; the introduction is overly condensed into a single paragraph, and the writing is characterized by terse sentences. A comprehensive rewrite is essential to enhance clarity, including the use of shorter sentences and multiple paragraphs to delineate contributions clearly.
    - The algorithms are inadequately formatted, rendering them challenging to read. Simplistic implementations need not be presented as formal algorithms; a descriptive narrative suffices.
    - The manuscript lacks descriptions of adequate motivation for the addressed problem, leading to insufficient context for readers. There is a disconnection between the problem statement and the proposed solutions.
    - Methodological explanations, particularly in Section 2, lack clarity and rigor, often employing casual language. Figure 2, in its current state, requires significant improvement to convey its intended message effectively.
- **Comparative Analysis**:
    - The comparisons with existing algorithms are insufficient. While the authors mention RDKit and ETKDG, there is a notable absence of comparisons with more recent methodologies, such as Torsional Diff [1].
    - Furthermore, standard evaluation metrics such as Precision and Recall are not reported, which are routinely used in evaluation of small molecule conformer generation models.
    - The rational for using PubChemQC as the target dataset is not convincingly justified over alternatives like GEOM-DRUGS[2], which are commonly used in this domain.

[1] Jing, Bowen, et al. "Torsional diffusion for molecular conformer generation." *Advances in Neural Information Processing Systems* 35 (2022): 24240-24253.

[2] Simon Axelrod and Rafael Gómez-Bombarelli. Geom, energy-annotated molecular conformations for property prediction and molecular generation. Scientific Data, 2022.

**Questions:**

1. Given that small molecules exhibit considerable conformational diversity, how does the model generate multiple conformers for a given structure? Are the baselines or evaluations accounting for this variability? Diffusion based generative models have explicit consideration for learning a conditional distribution of the conformers, which enables sampling, how does the current method handle this ?
2. Has the potential overlap with AlphaFold 3[3] been considered, as it appears to address similar objectives? Does this render the current approach somewhat redundant? Could the authors clearly articulate the main contributions and motivation of the paper under this context ?

[3] Abramson, Josh, et al. "Accurate structure prediction of biomolecular interactions with AlphaFold 3." *Nature* (2024): 1-3.

---

### Note · Authors · 2024-11-27

**Comment:**

We thank the reviewers for their valuable feedback. Unfortunately, changes needed to address the drawbacks of the manuscript include additional training runs, as well as extensive rewriting of the text. Therefore we are withdrawing this work to make necessary improvements.

**Withdrawal Confirmation:**

I have read and agree with the venue's withdrawal policy on behalf of myself and my co-authors.